# Gastrointestinal adverse events associated with tirzepatide: A bibliometric and pharmacovigilance analysis

Peng Shen[1,2], Meng-si Peng[1,2], Sun Jo Kim[1], Kyung-In Joung[3]*, Kwang Joon Kim[1]*

**1** College of Pharmacy, Chonnam National University, Gwangju, Republic of Korea, **2** School of Pharmaceutical Science, Wenzhou Medical University, Wenzhou, Zhejiang, China, **3** Division of Healthcare Science, College of Future Convergence, CHA University, Pocheon-si, Republic of Korea

* kjkim0901@jnu.ac.kr (KJK); jki0515@cha.ac.kr (KIJ)

## Abstract

### Background

This study examines gastrointestinal adverse events (GIAEs) associated with tirzepatide using bibliometric and pharmacovigilance analyses.

### Research design and methods

A bibliometric analysis of Web of Science data identified research trends in tirzepatide-related adverse events, while a pharmacovigilance analysis of FAERS data (Q2 2022–Q2 2024) assessed real-world GIAEs patterns. Disproportionality, time-to-onset, univariate, and comparative analyses were conducted to evaluate reporting odds ratios (RORs), onset timing, and subgroup differences.

### Results

Among 110 studies, cardiovascular outcomes predominated as the research focus. FAERS data showed that nausea (27.7%) and diarrhea (12.8%) were the most frequently reported events, whereas eructation and impaired gastric emptying had the highest disproportionality. GIAEs were more common in older adults, males, and patients receiving concomitant medications, and most occurred within 3 months (median onset: 16 days). Tirzepatide had a lower ROR for GIAEs than GLP-1 receptor agonists but a higher ROR than non-GLP-1 drugs, with a greater risk in patients with type 2 diabetes (T2DM) than in those using tirzepatide for weight loss.

### Conclusion

Tirzepatide is associated with an increased risk of GIAEs, particularly among patients with T2DM, males, older adults, and those using concomitant medications. FAERS-based real-world evidence complements clinical trial data and highlights the need for individualized patient monitoring and management strategies.

**Data availability statement:** The FAERS database is publicly available and anonymized, ensuring that no identifiable patient information is disclosed or used. It can be accessed publicly at: https://fis.fda.gov/extensions/FPD-QDE-FAERS/FPD-QDE-FAERS.html.

**Funding:** The author(s) received no specific funding for this work.

**Competing interests:** The authors have declared that no competing interests exist.

## Introduction

Type 2 diabetes mellitus (T2DM) is a common metabolic disorder characterized by chronic hyperglycemia resulting from progressive β-cell dysfunction and insulin resistance [1]. Affecting over 537 million adults globally and projected to reach 783 million by 2045 [2], T2DM requires innovative therapies targeting glycemic control and associated comorbidities, including obesity and cardiovascular diseases [3,4].

Tirzepatide, a first-in-class dual agonist of glucose-dependent insulinotropic polypeptide (GIP) and glucagon-like peptide-1 (GLP-1) receptors, represents a major advance in T2DM management [5]. Approved by the Food and Drug Administration (FDA) in 2022 for T2DM and in 2023 for chronic weight management, tirzepatide is more efficacious than selective GLP-1 receptor agonists, achieving greater HbA1c reductions and substantial weight loss [6–8]. Its dual mechanism, enhancing insulin secretion, suppressing glucagon, delaying gastric emptying, and promoting satiety, supports both glycemic control and weight management. However, tirzepatide use is often associated with gastrointestinal adverse events (GIAEs) [9,10], including nausea, vomiting, and diarrhea, which, although non-life-threatening, are highly prevalent and may compromise treatment adherence and outcomes.

Although in clinical trials, GIAEs are common adverse effects of tirzepatide [11], their strict inclusion criteria and controlled settings limit generalizability to diverse real-world populations. Existing real-world studies primarily focus on overall adverse events [12–16], often neglecting the specific features and clinical relevance of GIAEs. Since GIAEs are a leading cause of treatment discontinuation, their underrepresentation in prior research constitutes a critical knowledge gap. Moreover, the dual GIP and GLP-1 receptor agonism of tirzepatide may yield a distinct GIAE profile not fully captured in studies of selective GLP-1 receptor agonists. Recent research indicates that GLP-1 receptor agonists show age- and sex-related differences in gastric emptying [17], underscoring the need for targeted evaluation of tirzepatide-associated GIAEs across demographic and clinical subgroups. Understanding these differences is essential for optimizing treatment outcomes across patient populations.

To address these gaps, we conducted a dual-method study integrating bibliometric and pharmacovigilance analyses. The bibliometric analysis mapped the research landscape and identified key trends in tirzepatide-related adverse events. Concurrently, a pharmacovigilance analysis using the FDA Adverse Event Reporting System (FAERS) data [18,19] examined real-world differences in GIAEs based on sex, age, indication, and concomitant medication use. By integrating these complementary approaches, this study advances understanding of the safety profile of tirzepatide and provides novel insights into how patient-specific factors influence GIAEs.

## Methods

This study used a dual-method design integrating bibliometric and pharmacovigilance analyses (Fig 1). The bibliometric analysis mapped research trends and thematic evolution in tirzepatide-associated adverse events using the Web of Science (WoS) Core Collection. The pharmacovigilance analysis examined real-world GIAEs using FAERS data from Q2 2022 to Q2 2024.

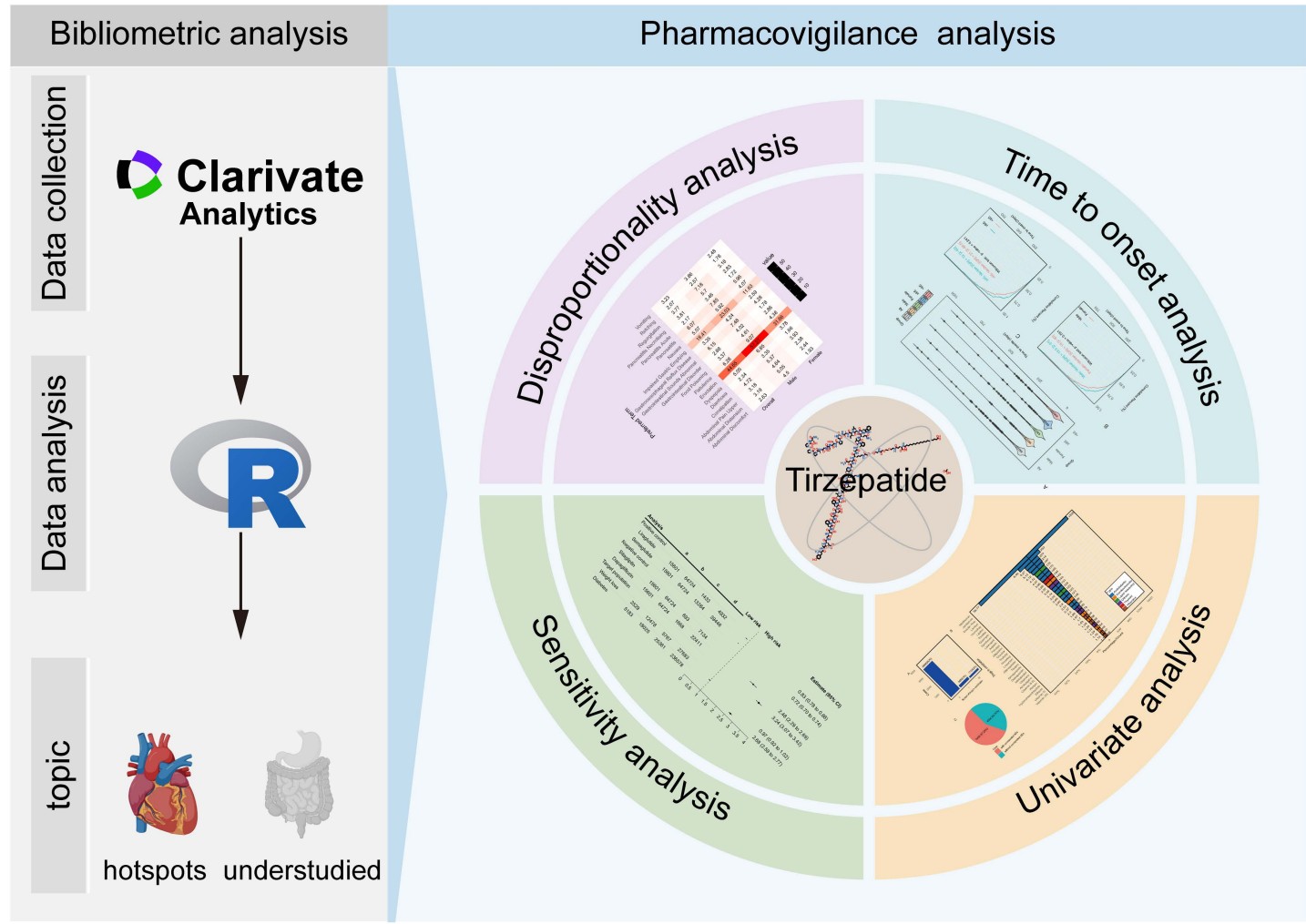

**Fig 1. Overview of the study design.**

## Bibliometric analysis

The WOS Core Collection was selected for its broad multidisciplinary coverage, which is essential for research on tirzepatide-associated adverse events spanning clinical medicine, pharmacology, biochemistry, and public health. Its organization, based on Bradford's and Garfield's laws, facilitates identification of core publications while minimizing omissions [20–24]. The primary search terms were "tirzepatide" and "adverse events," supplemented with synonyms to ensure comprehensive article retrieval. Data extracted from WoS on November 18, 2024, included 110 articles and reviews. Bibliometric analyses were performed in R (version 4.3.1) using the bibliometrix package [25] for data preprocessing, descriptive analysis, and mapping [26,27]. Authorship, country of origin, and keywords were extracted for each publication. Detailed bibliometric information, search strategy, and inclusion criteria are provided in the Supplementary Materials (S1 Table).

## Pharmacovigilance analysis

**Definition of cases and drugs of interest.** In the FAERS database, adverse events are coded using preferred terms (PTs) from the Medical Dictionary for Regulatory Activities (MedDRA) [28], organized into five hierarchical levels. PTs

uniquely describe medical concepts such as symptoms and diagnoses. Higher levels include "high-level terms" and "high-level group terms," which fall under system organ classes (SOCs) such as gastrointestinal disorders. This study focused on GIAEs within the SOC "gastrointestinal disorders," extracting 1,064 PTs from MedDRA version 27 (S2 Table). Only cases listing tirzepatide (S3 Table) as the primary suspect drug—the agent most likely to have caused the GIAEs—were included to minimize confounding.

**Data processing procedure.** The FAERS database contains seven interconnected datasets: DEMO (patient demographics), DRUG (drug details), INDI (drug indications), OUTC (patient outcomes), REAC (adverse events coded using MedDRA), PRSR (report sources), and THER (therapy data), all linked by a primary identifier. To remove duplicates, the most recent FDA_DT was retained when CASEID values matched, and the record with the higher PRIMARY_ID was kept when both CASEID and FDA_DT were identical, following FAERS user instructions [18,19,29,30]. Of 3,842,712 reported cases, 468,405 duplicates were removed, leaving 3,371,872 unique reports for analysis (S1 Fig).

## Descriptive analysis

Reports of tirzepatide-associated GIAEs included patient demographics (age, sex, and weight) and clinical details such as report type. Analyses were conducted in R (version 4.3.1), with data prepared using the *easyFAERS* package and figures generated using *ggplot2* [31]. Statistical significance was set at $p < 0.05$.

## Disproportionality analysis

Disproportionality analysis assesses potential associations between specific adverse events and drugs [32,33]. In this study, the reporting odds ratio (ROR) was used to evaluate tirzepatide-associated GIAEs, as in previous studies [34,35]. A drug–adverse event contingency table (S4 Table) was constructed to calculate the ROR and 95% confidence intervals (CIs) using the following formulas:

$$ROR = \frac{a/b}{c/d}$$

$$95\% \ CI = e^{\ln(ROR)\pm1.96\sqrt{\frac{1}{a}+\frac{1}{b}+\frac{1}{c}+\frac{1}{d}}}$$

A GIAE was considered significantly associated with tirzepatide if it had $\geq 3$ reports and the lower 95% CI limit exceeded 1.00 [36]. GI disorder PTs meeting these criteria were classified as tirzepatide-related [32,33]. The information component (IC) and proportional reporting ratio (PRR) were also calculated to supplement the ROR analysis [37].

## Time-to-onset analysis

Time-to-onset (TTO) was defined as the interval from the start of tirzepatide therapy to the occurrence of GIAEs. Cumulative distribution curves stratified by age and sex were used to visualize TTO, with statistical significance evaluated using the Wilcoxon rank-sum test [38]. The latency of tirzepatide-related GIAEs was further analyzed using the Weibull distribution model, characterized by the scale (α) and shape (β) parameters [39]. The scale parameter (α) represents the 63.2% quantile of events, while for β, a value $< 1$ indicates a decreasing hazard rate if the upper 95% CI limit is below 1.

## Univariate analysis

We conducted univariate logistic regression to examine the influence of selected variables on tirzepatide-associated GIAEs. Variables included age (<65 vs. ≥65 years), sex (male vs. female), clinical indication (diabetes vs. weight loss), and the number of concurrent medications. Odds ratios (OR) and 95% CIs were calculated to assess the strength of associations.

## Comparative analysis

To clarify the tirzepatide-associated GIAE profile, we conducted three comparative analyses: (1) Tirzepatide vs. GLP-1 receptor agonists (liraglutide, semaglutide) to isolate the effects of the dual tirzepatide GIP/GLP-1 mechanism; (2) tirzepatide vs. non-GLP-1 medications (sitagliptin, dapagliflozin) to differentiate GLP-1-specific effects from other pharmacological effects; (3) T2DM vs. obesity subgroups to evaluate outcomes in clinically relevant subpopulations.

## Ethics approval

The FAERS database is fully anonymized and publicly accessible (https://fis.fda.gov/extensions/FPD-QDE-FAERS/FPD-QDE-FAERS.html), containing no personally identifiable information.

This study was approved by the Institutional Review Board of Chonnam National University (No. 1040198–250108-HR-008–01), with informed consent waived because it involved no identifiable private data.

# Results

## Bibliometric analysis

Overall, 110 studies were identified, mostly from the United States and China (S1 Table), with the United States contributing 34 articles (30.9%) and China 16 (14.5%) (Fig 2A). Annual publications increased notably after FDA approval of tirzepatide in May 2022 (Fig 2B).

The thematic map (Fig 2C) categorizes tirzepatide-related adverse events into four quadrants based on development and relevance [40]. Cardiovascular outcomes were the most developed and central themes, reflecting research focus on the potential cardiovascular benefits of tirzepatide. Conversely, "weight," "therapy," and "cost prevalence" had lower density and centrality, indicating areas needing further study. Although GIAEs were central to this study, they did not appear as an independent theme in the bibliometric map.

## Pharmacovigilance analysis

**Descriptive analysis.** In total, 38,859 tirzepatide cases were analyzed, of which 9,490 (24.4%) reported GIAEs and 29,369 did not. Most individuals were under 65 years (54.4%) versus 11.1% aged ≥65 years. Females accounted for a higher portion of both GIAE cases (5,598, 59.0%) and non-GIAE (20,378, 69.4%) cases. Notably, among GIAE cases, males and those aged ≥65 years represented a substantial proportion relative to their overall dataset representation (Table 1).

## Disproportionality analysis

Analysis of tirzepatide-associated GIAE reports revealed that the most frequently reported GIAEs were nausea (4,323, 27.7%), diarrhea (1,999, 12.8%), vomiting (1,655, 10.6%), constipation (1,270, 8.1%), and upper abdominal pain (769, 4.9%) (S5 Table).

Tirzepatide was associated with a higher ROR for GIAEs (ROR: 2.82 [95% CI: 2.77–2.87]). Males had a higher ROR than females (3.44 vs. 2.17) and individuals ≥65 years had a higher ROR than those <65 years (2.78 vs. 1.91) (Fig 3A).

Venn diagrams illustrated GIAE distribution based on sex (Fig 3B) and age (Fig 3C). Tirzepatide was significantly associated with 20 gastrointestinal PTs across both sexes (ROR: 1.66–57.27), the highest being for Eructation (Fig 3D). Similarly, 17 PTs were significant across all age groups (ROR: 1.73–44.65), again with Eructation being the highest (Fig 3E). Some conditions exhibited demographic variations: faecaloma was significant only in females and individuals <65 years, while irritable bowel syndrome and frequent bowel movements were significant only in those ≥65 years (S6 Table). Results were consistent with IC and PRR analyses (S7–9 Table).

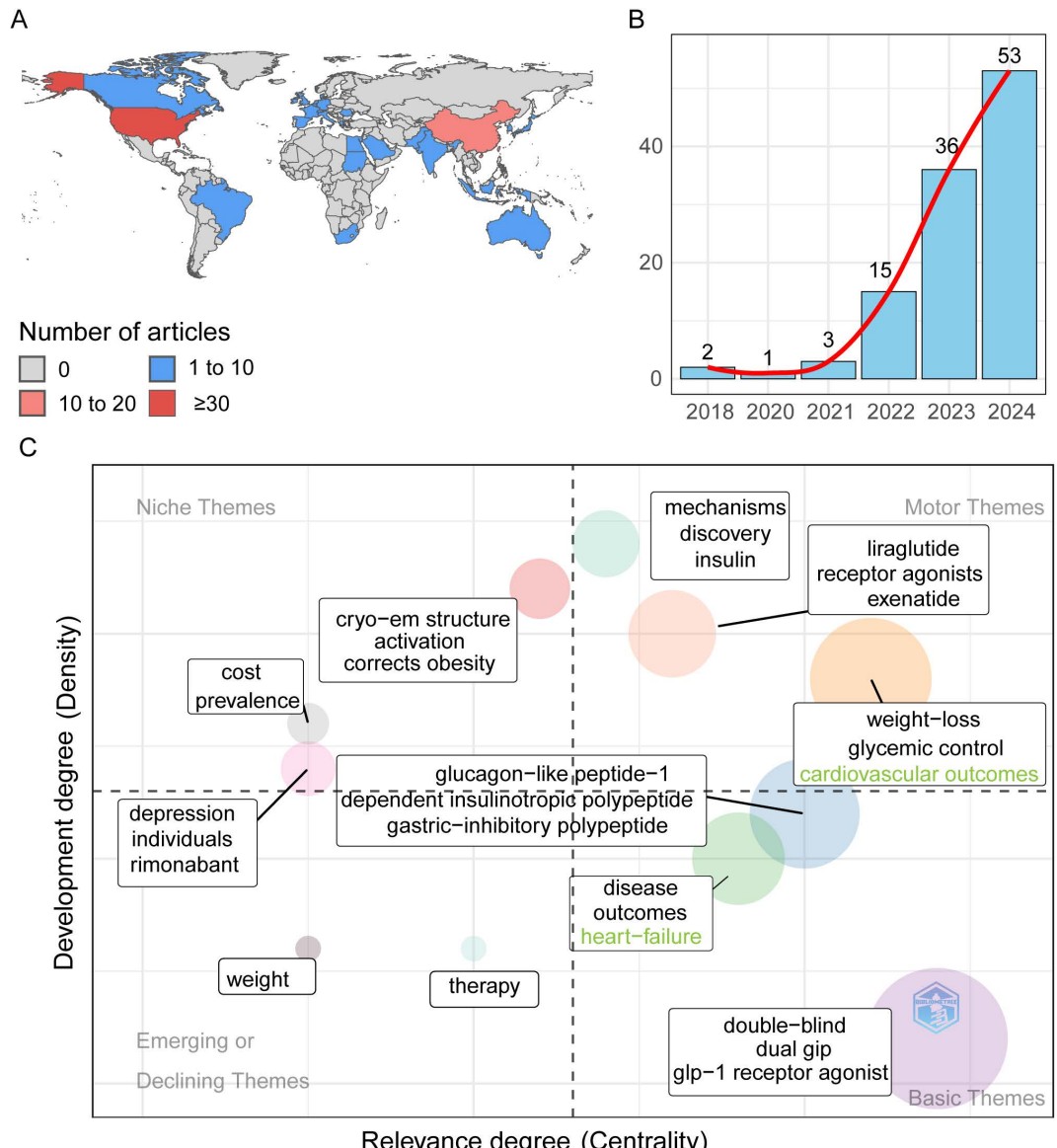

**Fig 2. Overview of the bibliometric analysis of tirzepatide-related adverse events. (A)** Scientific output by country. **(B)** Annual scientific production. **(C)** Thematic map of tirzepatide-related adverse events. Each theme is characterized by two metrics: density, indicating the degree of development, and centrality, reflecting cohesiveness within clusters and correlation across topics. Higher density denotes greater development, and higher centrality indicates greater relevance.

## Time-to-onset analysis

The temporal distribution of tirzepatide-associated GIAEs was consistent across the overall population and subgroups categorized by age and sex (Fig 4A). Over 80% of tirzepatide-related GIAEs occurred within the first 3 months of treatment, with a median onset of 16 days (interquartile range: 2–70 days). Onset timing was not significantly affected by sex (males: 13.5 vs. females: 18.0 days; $P = 0.531$; Fig 4B) or age (≥65 years: 12.0 vs. <65 years: 21.0 days; $P = 0.241$; Fig 4C). Shape parameters (β) and their 95% CI upper limits below 1 indicate an early failure pattern for tirzepatide-related GIAEs (S10 Table).

**Table 1. Characteristics of cases with tirzepatide-related gastrointestinal disorder sourced from the FAERS database.**

| Characteristics | non-GIAEs | GIAEs | Overall |
|---|---|---|---|
| Total | 29,369 | 9,490 | 38,859 |
| Sex | | | |
| Female | 20,378 (69.4%) | 5598 (59.0%) | 25,976 (66.8%) |
| Male | 5,805 (19.8%) | 1944 (20.5%) | 7,749 (19.9%) |
| Missing | 3,186 (10.8%) | 1948 (20.5%) | 5,134 (13.2%) |
| Weight (kg) | | | |
| <50 | 16 (0.1%) | 6 (0.1%) | 22 (0.1%) |
| >100 | 202 (0.7%) | 146 (1.5%) | 348 (0.9%) |
| 50–100 kg | 318 (1.1%) | 217 (2.3%) | 535 (1.4%) |
| Missing | 28,833 (98.2%) | 9,121 (96.1%) | 37,954 (97.7%) |
| Age (years) | | | |
| <65 | 17,286 (58.9%) | 3,846 (40.5%) | 21,132 (54.4%) |
| ≥65 | 3,211 (10.9%) | 1,112 (11.7%) | 4,323 (11.1%) |
| Missing | 8,872 (30.2%) | 4,532 (47.8%) | 13,404 (34.5%) |
| Reporter [a] | | | |
| Professionals | 1,733 (5.9%) | 624 (6.6%) | 2,357 (6.1%) |
| Non-professionals | 27,614 (94.0%) | 8,852 (93.3%) | 36,466 (93.8%) |
| Missing | 22 (0.1%) | 14 (0.1%) | 36 (0.1%) |
| Report country | | | |
| US | 28,917 (98.5%) | 9,325 (98.3%) | 38,242 (98.4%) |
| Non-US | 452 (1.5%) | 165(1.7%) | 617 (1.6%) |
| Report year | | | |
| 2022 | 2,136 (7.3%) | 981 (10.3%) | 3,117 (8.0%) |
| 2023 | 11,611 (39.5%) | 4,215 (44.4%) | 15,826 (40.7%) |
| 2024 | 15,622 (53.2%) | 4,294 (45.2%) | 19,916 (51.3%) |

[a]Professionals include reporters such as physicians and pharmacists; non-professionals include reporters such as consumers and lawyers.

Abbreviations: GIAEs, gastrointestinal adverse events; US, United States; FAERS; FDA Adverse Event Reporting System

## Factors associated with tirzepatide-related gastrointestinal adverse events

We analyzed tirzepatide use and found that 89.3% of patients used it alone, 6.3% with one additional drug, and 4.4% with two or more (Fig 5A). The most common co-administered drugs were metformin (Glucophage®), empagliflozin (Jardiance®), insulin (Novolog®), insulin glargine (Lantus®), and insulin lispro (Humalog®) (Fig 5B). Among 9,490 tirzepatide-related GIAE cases, 57.28% also experienced other adverse events (Fig 5C), most frequently in the categories "injury, poisoning, and procedures," "general disorders," and "metabolism and nutrition," (20–45% of cases) (Fig 5D). Patients using tirzepatide for diabetes had higher odds of reporting GIAEs (OR: 1.42) than those using it for weight management. The odds were also elevated for patients on one (OR: 1.36) or two or more concomitant drugs (OR: 1.30) versus monotherapy (Fig 5E).

## Comparative analysis

We evaluated tirzepatide-related GIAEs across drug comparisons and subgroups (Fig 6). Compared to other GLP-1 receptor agonists, tirzepatide had lower RORs: 0.83 (95% CI: 0.78–0.88) versus liraglutide and 0.72 (95% CI: 0.70–0.74) versus semaglutide. In contrast, higher RORs were observed when compared to non-GLP-1 medications: 2.48 (95% CI: 2.29–2.69) for sitagliptin and 3.24 (95% CI: 3.07–3.42) for dapagliflozin.

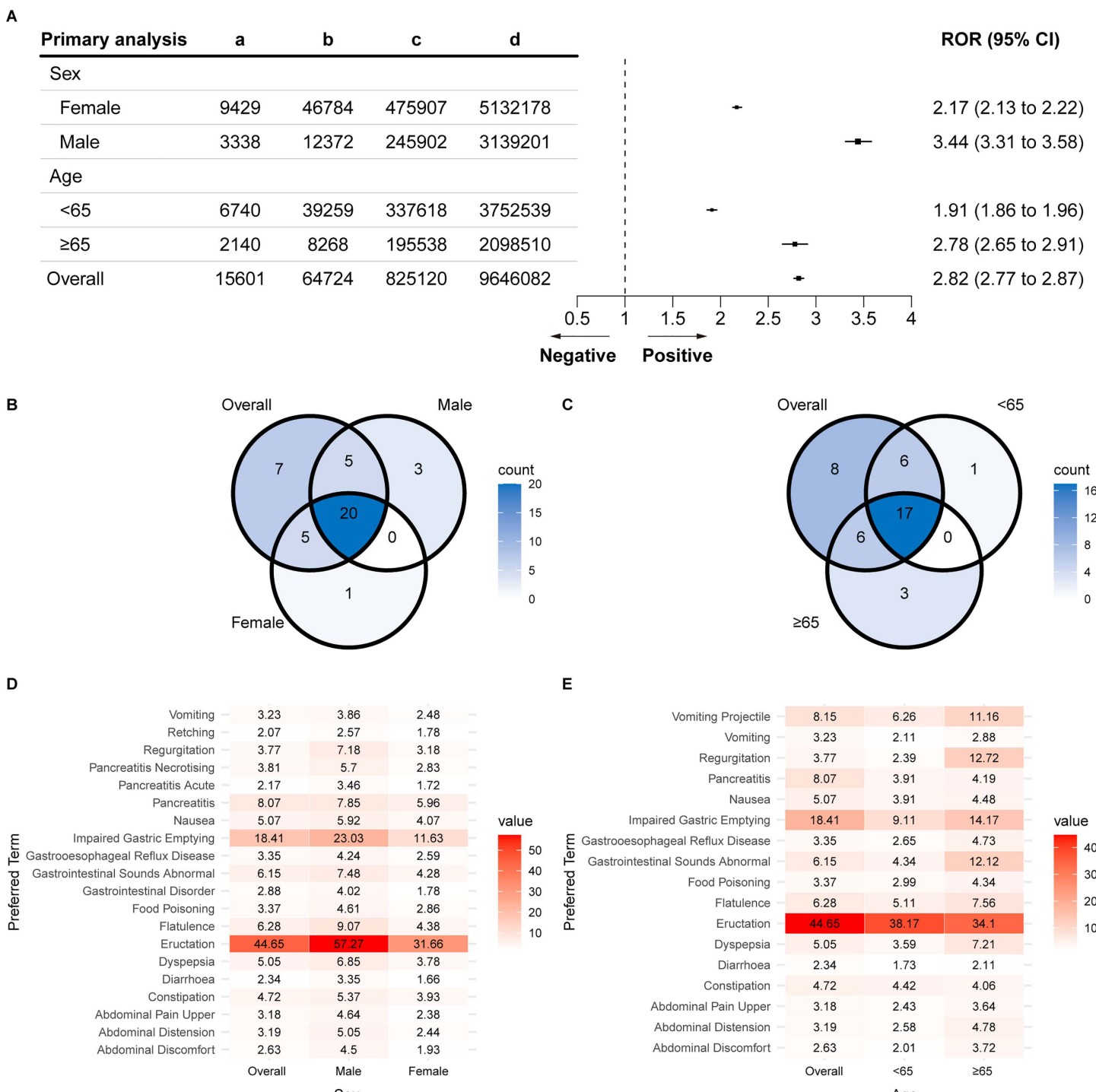

**Fig 3. RORs of tirzepatide-associated GIAEs. (A)** ROR at the SOC level. **(B)** Venn diagram of ROR based on sex at the PT level. **(C)** Venn diagram of ROR based on age at the PT level. **(D)** All positive ROR across sexes at the PT level. **(E)** All positive ROR across age groups at the PT level. Abbreviations: ROR, reporting odds ratio; GIAEs, gastrointestinal adverse events; SOC, system organ class; PT, preferred term.

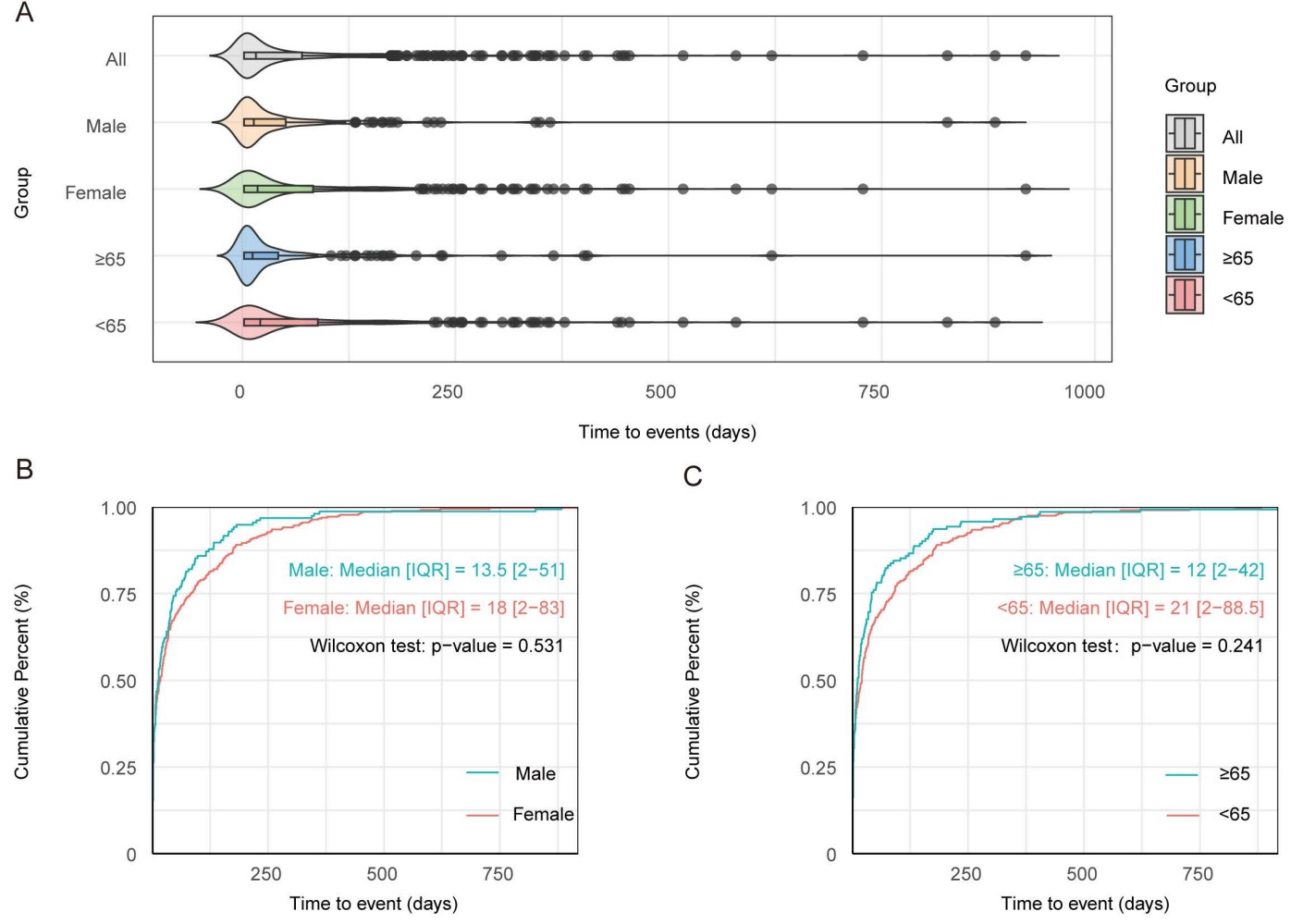

**Fig 4. Time-to-onset analysis. (A)** Violin plot of tirzepatide-related GIAE onset across age and sex subgroups. Cumulative distribution curves show tirzepatide-related GIAE onset stratified by **(B)** sex and **(C)** age. Statistical significance was assessed using the nonparametric Wilcoxon rank-sum test. Abbreviation: GIAEs, gastrointestinal adverse events.

Subgroup analyses showed elevated ROR for tirzepatide-related GIAEs in patients with T2DM (ROR: 2.68, 95% CI: 2.59–2.77) compared to those using tirzepatide for weight loss (ROR: 0.97, 95% CI: 0.92–1.02). These findings reveal distinct reporting patterns across drug classes and populations, clarifying the relative safety profile of tirzepatide.

## Discussion

Our study integrates bibliometric and pharmacovigilance analyses to evaluate tirzepatide-associated GIAEs. Bibliometric results show that, since FDA approval in 2022, research has largely focused on cardiovascular events, while GIAEs remain underrepresented. This emphasis reflects the high cardiovascular risk events in patients with T2DM—including myocardial infarction, stroke, and heart failure—and regulatory requirements for cardiovascular outcomes trials to ensure the safety of new antidiabetic drugs [41]. Clinically, antidiabetic therapies that control blood glucose and confer cardiovascular benefits are prioritized, particularly in high-risk patients with high cardiovascular risk, consistent with guidelines [42]. Although gastrointestinal effects are important for GLP-1 receptor agonists and cardiovascular outcomes, few studies

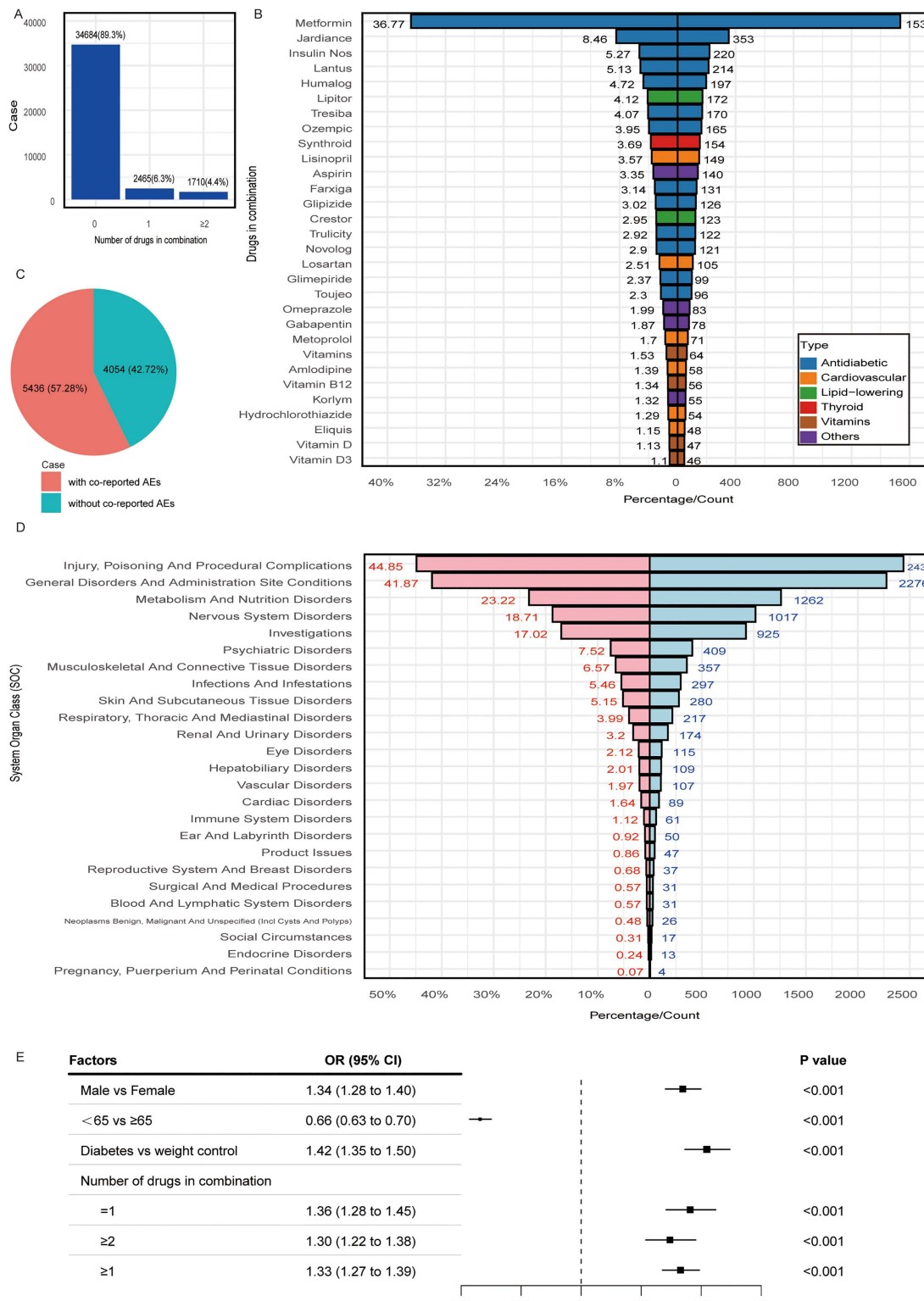

**Fig 5. Factors influencing tirzepatide-related GIAEs. (A)** Number of drugs used in combination. **(B)** Percentage of tirzepatide-related GIAEs with and without co-reported adverse events. **(C)** SOC statistics for PTs of co-reported adverse events, with counts and percentages of cases among tirzepatide-related GIAEs. **(D)** Top 30 co-administered drugs based on reported frequency. **(E)** Univariate logistic regression results for factors associated with tirzepatide-related GIAEs. Abbreviations: GIAEs, gastrointestinal adverse events; SOC, system organ class; PTs, preferred terms.

| Senstivity analysis | a | b | c | d | | ROR (95% CI) |
|---|---|---|---|---|---|---|
| Positive control | | | | | | |
| Liraglutide | 15601 | 64724 | 1433 | 4932 | | 0.83 (0.78 to 0.88) |
| Semaglutide | 15601 | 64724 | 13284 | 39446 | | 0.72 (0.70 to 0.74) |
| Negative control | | | | | | |
| Sitagliptin | 15601 | 64724 | 693 | 7134 | | 2.48 (2.29 to 2.69) |
| Dapagliflozin | 15601 | 64724 | 1668 | 22411 | | 3.24 (3.07 to 3.42) |
| Target population | | | | | | |
| Weight loss | 2529 | 12478 | 5767 | 27683 | | 0.97 (0.92 to 1.02) |
| Diabetes | 5183 | 18025 | 25361 | 236578 | | 2.68 (2.59 to 2.77) |

**Fig 6. Sensitivity analysis of RORs for tirzepatide-related GIAEs.** Abbreviations: ROR, reporting odds ratio; GIAEs, gastrointestinal adverse events.

specifically examine tirzepatide-related GIAEs, revealing a clear gap in the literature. This gap between literature focus and real-world pharmacovigilance underscores the value of integrating bibliometric and FAERS analyses to identify clinically relevant adverse events for further study and patient monitoring.

While clinical trials commonly report nausea, vomiting, and diarrhea as common GIAEs, FAERS data show disproportionate reporting of eructation and impaired gastric emptying. Our findings align with previous FAERS studies showing higher GIAE RORs among older adults and males [12]. We also observed increased GIAE risk with concomitant medications, supporting earlier reports [43] that GLP-1 receptor agonists carry varying gastrointestinal risks depending on co-administered drugs.

Unlike prior studies that primarily examined overall adverse events [12–16], this study provides a targeted pharmacovigilance analysis of tirzepatide-related GIAEs. These findings highlight the need for further research into risk mitigation and long-term safety, particularly in high-risk subgroups such as older adults and patients with pre-existing gastrointestinal disorders. Moreover, the TTO analysis revealed that most GIAEs occurred within the first 3 months of treatment, with a median onset of 16 days. This early-onset pattern suggests that proactive patient monitoring and gradual dose titration may be essential for improving treatment adherence and minimizing adverse effects.

The dual tirzepatide agonism of GLP-1 and GIP receptors modulates gastrointestinal function through multiple interconnected pathways. GLP-1 receptor activation slows gastric emptying by increasing pyloric tone and reducing antral contractility, thereby delaying nutrient transit from the stomach to the small intestine [44,45]. This effect likely contributes to commonly reported GIAEs, including early satiety, bloating, and nausea. In patients with T2DM, who often exhibit heterogeneous gastric motility, GLP-1 receptor agonists may further alter gastric emptying and symptom expression [44,45]. Moreover, older age and male sex may be associated with changes in autonomic regulation and gastrointestinal transit, predisposing some individuals to increased intragastric retention and gastrointestinal intolerance. Collectively, the combined GLP-1 and GIP agonism of tirzepatide may produce a distinct GIAE profile, underscoring the need for individualized treatment based on patient-specific metabolic and gastrointestinal characteristics.

Recent advances in multi-omics have enabled the coordinated integration of genomic, transcriptomic, proteomic, and metabolomic datasets into unified analytical frameworks. By interrogating multiple molecular layers simultaneously, these approaches uncover cross-scale regulatory networks and pathway interactions that are frequently obscured in single-omics analyses. In addition, their capacity to capture inter-individual molecular heterogeneity facilitates a more precise and mechanistically grounded characterization of drug–target–pathway relationships [46,47]. As these integrative strategies continue to evolve, they are increasingly being applied to adverse drug reactions research [48,49], providing deeper mechanistic insight and facilitating more refined pharmacovigilance practices within the framework of precision medicine.

This study leverages real-world FAERS pharmacovigilance data to identify tirzepatide-associated GIAEs beyond controlled clinical trial settings. The large dataset enhances generalizability by capturing diverse patient populations. Multiple analytical approaches, including disproportionality, TTO, and stratified subgroup analyses, were applied to comprehensively assess GIAE risk and temporal patterns. In addition, comparative analyses across drug classes and clinical indications were conducted to contextualize tirzepatide-associated GIAEs. These comparisons helped distinguish the effects of the dual GIP/GLP-1 agonism of tirzepatide from those of other GLP-1 receptor agonists and non-GLP-1 agents, offering deeper insight into its safety profile beyond signal detection.

However, this study has several limitations. First, it relied exclusively on the WoS Core Collection database, which may have excluded relevant studies indexed in other databases. Nevertheless, its broad coverage makes a substantial impact on observed trends unlikely. Second, as a spontaneous reporting system, FAERS is subject to underreporting, reporting bias, and duplicate records, potentially affecting data completeness and reliability [37,50]. Moreover, FAERS lacks detailed clinical information, including drug dosage, treatment duration, event severity, management strategies, comorbidities, hepatic and renal function status, and race. Collectively, these constraints limited assessment of dose–response relationships, temporal patterns, and GIAE severity, and precluded advanced analyses such as multivariate regression or propensity score matching.

Third, this analysis did not include a clinical priority assessment [51,52] because FAERS lacks sufficient mortality-related data for tirzepatide-associated GIAEs, potentially limiting clinical interpretability. Finally, as an observational study, causal relationships between tirzepatide and GIAEs cannot be established. To address these limitations, future studies should integrate FAERS data with other real-world data sources, such as electronic health records and prospective cohorts, to obtain more granular clinical information and enhance causal inference. Complementary approaches, including active surveillance systems, advanced signal detection methods, and mechanistic studies, may further validate pharmacovigilance findings. Clinically, careful interpretation of FAERS signals alongside randomized controlled trials and real-world cohort evidence is essential to enhance their relevance to patient care.

### Clinical Implications

Clinically, these findings emphasize the need for close monitoring during the first 3 months of tirzepatide treatment, when GIAEs mostly occur. Patient education on early gastrointestinal symptoms is essential. Gradual and individualized dose titration may help mitigate early-onset GIAEs, particularly in susceptible individuals. Special attention is warranted for older adults, males, patients using concomitant medications, and those with T2DM, who may be at higher risk of gastrointestinal intolerance.

### Conclusion

Tirzepatide is associated with an increased risk of GIAEs, particularly among patients with T2DM, males, older adults, and those using concomitant medications. FAERS-based real-world evidence complements clinical trial findings and highlights the need for individualized patient monitoring and management.

### Supporting information

**S1 Fig. The main steps in the processing of the FAERS database.**
(DOCX)

**S1 Table. Web of Science search strategy.**
(DOCX)

**S2 Table. MedDRA 27 search terms for gastrointestinal disorders.**
(DOCX)

**S3 Table. Generic and brand names of tirzepatide.**
(DOCX)

**S4 Table. Contingency table for ROR calculation.**
(DOCX)

**S5 Table. Number of GIAE cases in patients receiving tirzepatide treatment in FAERS (2022–2024).**
(DOCX)

**S6 Table. RORs of tirzepatide-associated GIAEs at the PT level.**
(DOCX)

**S7 Table. IC and PRR of tirzepatide-associated GIAEs at the SOC Level.**
(DOCX)

**S8 Table. IC of tirzepatide-associated GIAEs at the PT level.**
(DOCX)

**S9 Table. PRR tirzepatide-associated GIAEs at the PT level.**
(DOCX)

**S10 Table. Time-to-onset analysis of tirzepatide-related GIAEs.**
(DOCX)

## Acknowledgments

We thank the U.S. Food and Drug Administration for providing free access to the data used in this study.

## Author contributions

**Conceptualization:** Peng Shen, Kyung-In Joung, Kwang Joon Kim.

**Data curation:** Peng Shen, Meng-si Peng.

**Formal analysis:** Peng Shen, Meng-si Peng, Sun Jo Kim.

**Investigation:** Peng Shen.

**Methodology:** Peng Shen, Kyung-In Joung, Kwang Joon Kim.

**Supervision:** Kyung-In Joung, Kwang Joon Kim.

**Visualization:** Peng Shen.

**Writing – original draft:** Peng Shen.

**Writing – review & editing:** Sun Jo Kim, Kyung-In Joung, Kwang Joon Kim.

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
