## [Decision Letter · Decision Letter 0]

21 Apr 2025

Dear Dr. Kim,

Thank you for submitting your manuscript to PLOS ONE. After careful consideration, we feel that it has merit but does not fully meet PLOS ONE’s publication criteria as it currently stands. Therefore, we invite you to submit a revised version of the manuscript that addresses the points raised during the review process.

We look forward to receiving your revised manuscript.

Kind regards,

Jingyu Wang

Academic Editor

PLOS ONE

Journal Requirements:

Reviewers' comments:

Reviewer's Responses to Questions

**Comments to the Author**

1. Is the manuscript technically sound, and do the data support the conclusions?

Reviewer #1: Yes

Reviewer #2: Partly

2. Has the statistical analysis been performed appropriately and rigorously?

Reviewer #1: Yes

Reviewer #2: Yes

3. Have the authors made all data underlying the findings in their manuscript fully available?

Reviewer #1: Yes

Reviewer #2: Yes

4. Is the manuscript presented in an intelligible fashion and written in standard English?

Reviewer #1: Yes

Reviewer #2: Yes

Reviewer #1: The study was well conducted, appropriately designed, and well explained.

The chosen topic is of great scientific interest.

The authors present the results impartially. The strengths and limitations of the manuscript are adequately reported.

The tables and figures correspond to the presented results and are well addressed in the discussion.

The bibliography is up-to-date, although somewhat limited.

Reviewer #2: Thank you for the opportunity to review this manuscript. It's well written and addresses an important issue. However, in my analysis, it raises some concerns, that need to be addressed by the authors.

First of all, the manuscript actual focus on gastrointestinal adverse events and not gastrointestinal disorders. Even though that was the selected SOC in the pharmacovigilance analysis. I recommend a change in the title.

The Introduction does not support the relevance and novelty of this study. Lines 49-50, the authors state "... it's use is often associated with gastrointestinal adverse events, such as nausea, vomiting and diarrhea... ", which were the main adverse events also found in this analysis. The fact that other studies using real-word data didn't focus specifically in gastrointestinal adverse events is not justification enough. While studying overall adverse events, if gastrointestinal adverse events were that significant and/or relevant would not appear on the results of those studies?

Other relevant aspect in the Introducion is that no mention is made to the use of tirzepatide in obesity. Is it approved for the clinical use? Therefore, there's no sufficient background for the comparative analyses of T2DM patients vs obese patients.

In the methods sections, some questions arise that need clarification. First of all, why the single use of Web of Science for the bibliometric analysis? Doesn't that limits the scope of the analysis?

Secondly, the definition of cases and drugs of interest using a System Organ Class leads to too many Preferred Terms, which may disperse the results. However, that was the author's choice. My question is, if the study included cases where tirzepatide was the primary suspect drug, where the data for the comparative analyses came from? From FAERS also? Using what criteria?

Considering the results analysis and discussion, it lacks a merge between the bibliometric analysis and the pharmacovigilance analysis. Also, data on adverse events are of a different nature between those two sources. Concepts of association, causation are confused in this section.

Last comment is on the first sentence in the Discussion. "Our bibliometric analysis revealed a sharp increase in research activity on tirzepatide-associated gastrointestinal adverse events (...)". I don't think it did, because the authors also state in the results that "(...) did not appear as an independent theme in the bibliometric map", referring to gastrointestinal adverse events.

**Do you want your identity to be public for this peer review?** For information about this choice, including consent withdrawal, please see our For information about this choice, including consent withdrawal, please see our Privacy Policy .

Reviewer #1: No

Reviewer #2: **Yes:** André Filipe Ferreira CoelhoAndré Filipe Ferreira Coelho

While revising your submission, please upload your figure files to the Preflight Analysis and Conversion Engine (PACE) digital diagnostic tool, https://pacev2.apexcovantage.com/ . PACE helps ensure that figures meet PLOS requirements. To use PACE, you must first register as a user. Registration is free. Then, login and navigate to the UPLOAD tab, where you will find detailed instructions on how to use the tool. If you encounter any issues or have any questions when using PACE, please email PLOS at . PACE helps ensure that figures meet PLOS requirements. To use PACE, you must first register as a user. Registration is free. Then, login and navigate to the UPLOAD tab, where you will find detailed instructions on how to use the tool. If you encounter any issues or have any questions when using PACE, please email PLOS at figures@plos.org . Please note that Supporting Information files do not need this step.. Please note that Supporting Information files do not need this step.

---

## [Author Response · Author response to Decision Letter 1]

5 Jun 2025

Response to Editor and Reviewers

Dear Editor,

We would like to express our sincere gratitude to the reviewers for their insightful comments and constructive suggestions, which have significantly contributed to improving the quality and clarity of our manuscript.

We have carefully revised the manuscript in response to the reviewers’ feedback and are resubmitting it for your kind consideration. All amendments have been highlighted in yellow in the revised version. A detailed, point-by-point response to each comment is provided below this letter.

We hope that the revised manuscript meets the requirements for publication in your journal. We look forward to your favorable response.

With best regards,

Corresponding Author:

Kwang Joon Kim, PhD

College of Pharmacy, Chonnam National University, Gwangju 61186, Republic of Korea

Phone: 82 625302929

Fax: 82 625302949

Email: kjkim0901@jnu.ac.kr

Co-corresponding Author

Kyung-In Joung, PnD

School of AI Healthcare, College of Integrated Health Science, CHA University, Pocheon-si 11160, Republic of Korea

Phone: 82 318509087

Fax: 82 315439917

Email: jki0515@cha.ac.kr

We have provided our detailed responses to the reviewers’ comments and suggestions as follows.

Reviewer #1:

Comment 1: The study was well conducted, appropriately designed, and well explained. The chosen topic is of great scientific interest. The authors present the results impartially. The strengths and limitations of the manuscript are adequately reported. The tables and figures correspond to the presented results and are well addressed in the discussion. The bibliography is up-to-date, although somewhat limited.

Response 1: We are deeply grateful for your insightful comments and the time you devoted to reviewing our manuscript. Your careful assessment and constructive suggestions have greatly contributed to improving the overall quality of our work.

Reviewer #2:

Thank you for the opportunity to review this manuscript. It's well written and addresses an important issue. However, in my analysis, it raises some concerns, that need to be addressed by the authors.

Comment 1: First of all, the manuscript actual focus on gastrointestinal adverse events and not gastrointestinal disorders. Even though that was the selected SOC in the pharmacovigilance analysis. I recommend a change in the title.

Response 1: Thank you very much for this important and constructive comment. To better align with the main focus of the study, we have revised the manuscript title as follows: "Gastrointestinal adverse events associated with tirzepatide: a bibliometric and pharmacovigilance analysis"(line 1)

Comment 2: The Introduction does not support the relevance and novelty of this study. Lines 49-50, the authors state "... it's use is often associated with gastrointestinal adverse events, such as nausea, vomiting and diarrhea... ", which were the main adverse events also found in this analysis. The fact that other studies using real-word data didn't focus specifically in gastrointestinal adverse events is not justification enough. While studying overall adverse events, if gastrointestinal adverse events were that significant and/or relevant would not appear on the results of those studies?

Response 2: Thank you very much for this important and constructive comment. We agree that simply stating that previous studies did not focus on gastrointestinal adverse events (GIAEs) is not sufficient to establish the novelty or necessity of our study. We have therefore revised the Introduction to more clearly articulate why a focused investigation on GIAEs is clinically and scientifically justified.

First, although GIAEs such as nausea, vomiting, and diarrhea have frequently been reported with tirzepatide, they have often been analyzed in aggregated safety outcomes without in-depth exploration. These events, though non-fatal, are among the most common causes of treatment discontinuation and reduced adherence, which has direct implications for real-world therapeutic effectiveness. Despite their frequency, most real-world studies have treated GIAEs as background noise rather than clinically significant events warranting focused evaluation.

Second, tirzepatide has a unique mechanism of action as a dual GIP/GLP-1 receptor agonist. This distinguishes it from previously studied GLP-1 receptor agonists and may lead to a distinct profile of gastrointestinal effects. For example, eructation and gastric emptying delay—highly disproportionate events in our study—have not been systematically characterized in earlier research. This pharmacological distinction provides a mechanistic rationale for examining tirzepatide-specific GIAEs in greater detail.

Third, recent research has shown that GLP-1 receptor agonists exhibit sex- and age-related differences in gastric emptying [Ref. 17], and such variability has not been explored in tirzepatide. Our study aims to fill this gap by assessing GIAE risk across demographic (age, sex), clinical (T2DM vs. obesity), and pharmacological (concomitant medication) subgroups.

Therefore, the novelty of our study lies not merely in isolating GIAEs, but in providing a stratified, real-world safety assessment using complementary approaches: bibliometric mapping to highlight research gaps, and pharmacovigilance analysis to explore subgroup differences. This integrative design enhances our understanding of tirzepatide’s safety profile and addresses the underrepresentation of GIAEs in the literature despite their prevalence in spontaneous reports.

We have reflected these revisions in the Introduction section (lines 53–64) to ensure that the rationale, relevance, and novelty of our research are clearly communicated.

We sincerely appreciate again for your highlighting this critical point, which helped us improve both the clarity and strength of our argument.

Comment 3: Other relevant aspect in the Introducion is that no mention is made to the use of tirzepatide in obesity. Is it approved for the clinical use? Therefore, there's no sufficient background for the comparative analyses of T2DM patients vs obese patients.

Response 3: We appreciate your insightful comment. You rightly pointed out that our original manuscript did not address the use of tirzepatide in obesity or its current approval status. To provide a more comprehensive background and to strengthen the basis for comparing its use in patients with type 2 diabetes mellitus (T2DM) and those with obesity, we have added relevant information regarding the clinical approval of tirzepatide for obesity and its potential therapeutic role in this population (lines 44–50).

Comment 4: In the methods sections, some questions arise that need clarification. First of all, why the single use of Web of Science for the bibliometric analysis? Doesn't that limits the scope of the analysis?

Response 4:

First, the topic of "adverse events associated with tirzepatide" spans multiple disciplines—including Clinical Medicine, Pharmacology, Biochemistry, and Public Health—which necessitates the use of a comprehensive and integrated database to ensure the inclusion of relevant literature across various fields.

Second, the WOS Core Collection includes the Science Citation Index Expanded (SCIE), a flagship citation index known for its inclusion of high-quality, peer-reviewed journals.

Third, the database is structured in accordance with Bradford’s Law and Garfield’s Law, enhancing the likelihood that core publications are captured while minimizing the risk of omissions. The exclusive use of the WOS Core Collection is further supported by recent high-quality studies that have employed this dataset alone [10–12].

Nonetheless, we acknowledge the inherent limitation of relying on a single database, which may have resulted in the exclusion of relevant studies from other sources. To address this concern, we included the following statement in the Discussion section (lines 328–332):

" Firstly, the study relied exclusively on the WOS Core Collection database as its source of research articles, which may have excluded valuable studies published in other databases. However, given the extensive coverage of the WOS Core Collection across most studies, this limitation is unlikely to have significantly impacted the observed trends."

Comment 5: Secondly, the definition of cases and drugs of interest using a System Organ Class leads to too many Preferred Terms, which may disperse the results. However, that was the author's choice. My question is, if the study included cases where tirzepatide was the primary suspect drug, where the data for the comparative analyses came from? From FAERS also? Using what criteria?

Response 5: Thank you for your thoughtful question. We appreciate the opportunity to clarify the criteria employed in our comparative analyses.

As described in the Methods section (lines 100–104), we utilized the FDA Adverse Event Reporting System (FAERS) database, which comprises seven interlinked datasets: DEMO (demographics), DRUG (drug data), INDI (indications), OUTC (outcomes), REAC (adverse events coded via MedDRA), PRSR (report source), and THER (therapy dates). In the DRUG file, each drug listed in an adverse event report is assigned a specific role—primary suspect (PS), secondary suspect, concomitant, or interacting drug.

To ensure the accuracy of causal attribution, we restricted our analyses to cases in which tirzepatide was designated as the primary suspect drug. This criterion was consistently applied across all stages of analysis, including the selection of gastrointestinal adverse event (GIAE) cases and all subsequent comparative evaluations (e.g., comparisons with other GLP-1 receptor agonists, non-GLP-1 agents, or stratified analyses by indication, sex, or age). Reports in which tirzepatide was classified as a secondary suspect, concomitant, or interacting agent were excluded, as these designations offer a lower level of causal inference reliability [13–15].

We hope this explanation clarifies the methodological rigor underlying our study. We are grateful for your valuable question, which prompted us to improve the transparency and precision of our description.

Comment 6: Considering the results analysis and discussion, it lacks a merge between the bibliometric analysis and the pharmacovigilance analysis. Also, data on adverse events are of a different nature between those two sources. Concepts of association, causation are confused in this section.

Response 6: Thank you for your insightful comments regarding the integration of findings and the distinction between association and causation. We agree that the initial version of the Discussion section did not sufficiently synthesize the results from the bibliometric and pharmacovigilance analyses.

In response, we revised the Discussion section (lines 263–275) to explicitly integrate the findings from both components of the study. We emphasized how the bibliometric analysis revealed a gap in the literature concerning detailed evaluation of gastrointestinal adverse events (GIAEs), a gap we sought to address using real-world pharmacovigilance data.

Regarding your concern about the potential confusion between association and causation, we have been careful throughout the manuscript to use terminology that clearly distinguishes between the two concepts. Specific sections where this distinction is maintained include lines 23, 189–192, 233, 280–281, 284, and 310. We ensured that the language used avoids causal implication and accurately reflects the observational nature of the data.

We sincerely appreciate your thoughtful feedback, which has significantly contributed to improving the conceptual clarity of our manuscript.

Comment 7: Last comment is on the first sentence in the Discussion. "Our bibliometric analysis revealed a sharp increase in research activity on tirzepatide-associated gastrointestinal adverse events (...)". I don't think it did, because the authors also state in the results that "(...) did not appear as an independent theme in the bibliometric map", referring to gastrointestinal adverse events.

Response 7: Thank you for your valuable observation. We understand your concern and appreciate the opportunity to clarify the intended meaning of our statement.

Our intention was not to imply that gastrointestinal adverse events (GIAEs) had emerged as a major theme in the bibliometric analysis. Rather, as elaborated more clearly in the revised discussion section, we observed that although research interest in tirzepatide’s overall safety profile has increased substantially following FDA approval, GIAEs were not identified as a distinct thematic cluster in the bibliometric mapping. This observed gap in the literature further reinforces the rationale for our integrated approach.

We appreciate your comment, which enabled us to clarify this important point in the manuscript.

We hope that our integrated approach not only contributes to the growing understanding of tirzepatide’s safety profile but also highlights the importance of patient-specific evaluation in future pharmacovigilance research. We sincerely thank the reviewers for their thoughtful feedback, which has greatly improved the clarity and focus of our work.

References

1. FDA. Available from: https://www.accessdata.fda.gov/scripts/cder/daf/index.cfm?event=overview.process&ApplNo=217806.

2. FAD. Available from: https://www.accessdata.fda.gov/scripts/cder/daf/index.cfm?event=overview.process&ApplNo=217806.

3. Aronne LJ, Sattar N, Horn DB, Bays HE, Wharton S, Lin WY, et al. Continued Treatment With Tirzepatide for Maintenance of Weight Reduction in Adults With Obesity: The SURMOUNT-4 Randomized Clinical Trial. Jama. 2024;331(1):38-48. doi: 10.1001/jama.2023.24945. PubMed PMID: 38078870; PubMed Central PMCID: PMCPMC10714284.

4. Garvey WT, Frias JP, Jastreboff AM, le Roux CW, Sattar N, Aizenberg D, et al. Tirzepatide once weekly for the treatment of obesity in people with type 2 diabetes (SURMOUNT-2): a double-blind, randomised, multicentre, placebo-controlled, phase 3 trial. Lancet. 2023;402(10402):613-26. Epub 20230626. doi: 10.1016/s0140-6736(23)01200-x. PubMed PMID: 37385275.

5. Caruso I, Di Gioia L, Di Molfetta S, Caporusso M, Cignarelli A, Sorice GP, et al. The real-world safety profile of tirzepatide: pharmacovigilance analysis of the FDA Adverse Event Reporting System (FAERS) database. J Endocrinol Invest. 2024;47(11):2671-8. Epub 20240814. doi: 10.1007/s40618-024-02441-z. PubMed PMID: 39141075; PubMed Central PMCID: PMCPMC11473560.

6. Ou Y, Cui Z, Lou S, Zhu C, Chen J, Zhou L, et al. Analysis of tirzepatide in the US FDA adverse event reporting system (FAERS): a focus on overall patient population and sex-specific subgroups. Front Pharmacol. 2024;15:1463657. Epub 20241106. doi: 10.3389/fphar.2024.1463657. PubMed PMID: 39568578; PubMed Central PMCID: PMCPMC11576270.

7. Li J, Xie J, Han Y, Zhang W, Wang Y, Jiang Z. A real-world disproportionality analysis of tirzepatide-related adverse events based on the FDA Adverse Event Reporting System (FAERS) database. Endocr J. 2024. Epub 20241127. doi: 10.1507/endocrj.EJ24-0286. PubMed PMID: 39603650.

8. Liu L. A real-world data analysis of tirzepatide in the FDA adverse event reporting system (FAERS) database. Front Pharmacol. 2024;15:1397029. Epub 20240607. doi: 10.3389/fphar.2024.1397029. PubMed PMID: 38910884; PubMed Central PMCID: PMCPMC11190169.

9. Huo Y, Ma M, Liao X. Data mining study on adverse events of tirzepatide based on FAERS database. Expert Opin Drug Saf. 2024:1-9. Epub 20240715. doi: 10.1080/14740338.2024.2376686. PubMed PMID: 39007672.

10. Zhang L, Zheng H, Jiang ST, Liu YG, Zhang T, Zhang JW, et al. Worldwide research trends on tumor burden and immunotherapy: a bibliometric analysis. Int J Surg. 2024;110(3):1699-710. Epub 20240301. doi: 10.1097/js9.0000000000001022. PubMed PMID: 38181123; PubMed Central PMCID: PMCPMC10942200.

11. Wu T, Wu Y, Nie K, Yan J, Chen Y, Wang S, et al. Bibliometric analysis and global trends in uterus transplantation. Int J Surg. 2024;110(8):4932-46. Epub 20240801. doi: 10.1097/js9.0000000000001470. PubMed PMID: 38626445; PubMed Central PMCID: PMCPMC11326002.

12. Wang J, Zhao W, Zhang Z, Liu X, Xie T, Wang L, et al

---

## [Decision Letter · Decision Letter 1]

12 Aug 2025

Dear Dr. Kim,

ACADEMIC EDITOR:

1. Regarding the selection of the WOSCC core collection, it is essential that you elaborate on the reasons for choosing this database. You may refer to the following articles to support your explanation: PMID: 38229552, 38913439, 40104597, and 38345042.

2. For the analysis of FAERS, a clinical priority assessment should be conducted, and you can consult PMID: 39291217 and 39972562 for relevant methods and perspectives. In addition, when introducing FAERS, it is recommended to incorporate the following literatures: PMID: 40349689 and the article available at https://spj.science.org/doi/10.34133/hds.0325 to enhance the comprehensiveness and authority of your introduction.

We look forward to receiving your revised manuscript.

Kind regards,

Jingyu Wang

Academic Editor

PLOS ONE

Journal Requirements:

Additional Editor Comments:

Regarding the selection of the WOSCC, it is essential that you elaborate on the reasons for choosing this database. You may refer to the following articles to support your explanation: PMID: 38229552, 38913439, 40104597, and 38345042.

For the analysis of FAERS, a clinical priority assessment should be conducted, and you can consult PMID: 39291217 and 39972562 for relevant methods and perspectives. In addition, when introducing FAERS, it is recommended to incorporate the following literatures: PMID: 40349689 and the article available at https://spj.science.org/doi/10.34133/hds.0325 to enhance the comprehensiveness and authority of your introduction.

Reviewers' comments:

Reviewer's Responses to Questions

**Comments to the Author**

Reviewer #1: All comments have been addressed

Reviewer #3: (No Response)

2. Is the manuscript technically sound, and do the data support the conclusions?

Reviewer #1: Yes

Reviewer #3: Partly

3. Has the statistical analysis been performed appropriately and rigorously?

Reviewer #1: Yes

Reviewer #3: (No Response)

4. Have the authors made all data underlying the findings in their manuscript fully available?

Reviewer #1: Yes

Reviewer #3: (No Response)

5. Is the manuscript presented in an intelligible fashion and written in standard English?

Reviewer #1: Yes

Reviewer #3: (No Response)

Reviewer #1: The authors have improved the text and the manuscript has been changed as suggested. It is ready for publication.

Reviewer #3: This manuscript employs bibliometric analysis and pharmacovigilance methods to systematically evaluate research trends and real-world safety characteristics of tirzepatide-related gastrointestinal adverse events. The authors searched 110 relevant studies through the Web of Science database and analyzed 38,859 tirzepatide use cases based on the FAERS database, of which 9,490 cases reported gastrointestinal adverse events. The study found that nausea and diarrhea were the most frequently reported adverse events, while belching and gastric emptying impairment showed the highest reporting odds ratios. This research provides valuable reference information for safety monitoring of tirzepatide in clinical practice. However, the manuscript still has the following areas that need improvement regarding the rigor of research methods, depth of data analysis, and interpretation of clinical significance.

Major Comments

1. The authors should provide more detailed search strategies and screening flowcharts in the bibliometric analysis section. The current methodological description is overly simplified and lacks standardized systematic review processes. Particularly regarding keyword selection and database limitations, the authors could consider expanding the search scope to include other important biomedical databases such as PubMed and Embase to ensure comprehensive literature retrieval. Additionally, the authors should provide detailed descriptions of inclusion and exclusion criteria for literature screening and present inter-rater agreement results from two independent reviewers.

2. In the FAERS data analysis, the authors should consider adopting more rigorous statistical methods to address potential confounding factors. While the current disproportionality analysis employed reporting odds ratios, it did not consider other algorithms such as IC method, PRR method, etc. Furthermore, the authors should also incorporate the VigiBase database for additional supplementary analysis (doi: 10.1016/j.eclinm.2024.102684).

3. The authors need to further explore the dose-response relationship and time-dependent characteristics of tirzepatide gastrointestinal adverse events. The current analysis primarily focuses on the frequency and distribution of adverse events but lacks in-depth analysis of event severity, duration, and clinical management strategies. Recent molecular mechanism research on adverse drug reactions shows that gastrointestinal adverse events are often closely related to drug target specificity and signal transduction pathways. The authors should combine tirzepatide's dual GIP/GLP-1 receptor agonist mechanism to deeply analyze the immune microenvironmental basis of its gastrointestinal effects (PMID: 35331128; 37691196; doi: 10.1002/mdr2.70001; 10.1002/mdr2.70007).

4. Although the authors conducted subgroup analyses based on gender, age, and indications, they should further consider the impact of other important clinical variables on gastrointestinal adverse event incidence, such as comorbidities, hepatic and renal function status, and racial differences. Particularly in elderly patient populations, polypharmacy and organ function decline may significantly affect drug safety characteristics. The authors could employ multivariate regression analysis or propensity score matching methods to more accurately assess the impact of these confounding factors.

5. The two analytical components in the current study are relatively independent, lacking deep integration. The authors should establish correlation analyses between literature research hotspots and real-world adverse event reports, exploring differences between academic research focus and actual safety issues occurring in clinical practice. This integrated analysis is of significant importance for guiding future research directions and clinical monitoring priorities. With the widespread application of multi-omics technologies in drug discovery, prediction and mechanistic research of adverse drug reactions also require more systematic methodological support (doi: 10.1016/j.cpan.2024.12.001).

6. The authors should more comprehensively discuss the inherent limitations of the FAERS database, including reporting bias, missing information, limitations of causal inference, and propose corresponding solutions or explanatory approaches. Additionally, they need to explain how to overcome these limitations in actual clinical applications to improve the practical utility of research results.

**Do you want your identity to be public for this peer review?** For information about this choice, including consent withdrawal, please see our For information about this choice, including consent withdrawal, please see our Privacy Policy .

Reviewer #1: No

Reviewer #3: No

While revising your submission, please upload your figure files to the Preflight Analysis and Conversion Engine (PACE) digital diagnostic tool, https://pacev2.apexcovantage.com/ . PACE helps ensure that figures meet PLOS requirements. To use PACE, you must first register as a user. Registration is free. Then, login and navigate to the UPLOAD tab, where you will find detailed instructions on how to use the tool. If you encounter any issues or have any questions when using PACE, please email PLOS at . PACE helps ensure that figures meet PLOS requirements. To use PACE, you must first register as a user. Registration is free. Then, login and navigate to the UPLOAD tab, where you will find detailed instructions on how to use the tool. If you encounter any issues or have any questions when using PACE, please email PLOS at figures@plos.org . Please note that Supporting Information files do not need this step.

---

## [Author Response · Author response to Decision Letter 2]

15 Sep 2025

We have provided our detailed responses to the reviewers’ comments and suggestions as follows.

Editor #1:

Comment 1: Regarding the selection of the WOSCC core collection, it is essential that you elaborate on the reasons for choosing this database. You may refer to the following articles to support your explanation: PMID: 38229552, 38913439, 40104597, and 38345042.

Response 1:

We thank the Editor for this constructive comment. In the revised manuscript, we have expanded the rationale for selecting the Web of Science Core Collection (WoSCC). Specifically, we emphasized its multidisciplinary coverage (e.g., Clinical Medicine, Pharmacology, Biochemistry, and Public Health) and its foundation on Bradford’s Law and Garfield’s Law, which enhances retrieval of core publications. We also cited recent bibliometric studies (PMID: 38229552, 38913439, 40104597, and 38345042) that employed WoSCC as the sole data source, thereby supporting the validity of our methodological choice. The corresponding revisions can be found in the Methods section (lines 78-82).

Comment 2: For the analysis of FAERS, a clinical priority assessment should be conducted, and you can consult PMID: 39291217 and 39972562 for relevant methods and perspectives. In addition, when introducing FAERS, it is recommended to incorporate the following literatures: PMID: 40349689 and the article available at https://spj.science.org/doi/10.34133/hds.0325 to enhance the comprehensiveness and authority of your introduction.

Response 2：We sincerely thank the editor for this insightful suggestion. In the revised manuscript, we have expanded the description of FAERS by incorporating the recommended references (PMID: 40349689 and https://spj.science.org/doi/10.34133/hds.0325) to enhance the comprehensiveness and authority of the introduction (line 69).

With regard to the clinical prioritization evaluation, we fully agree on its importance in strengthening the interpretation of FAERS analyses (PMID: 39291217; 39972562). The approach described in these references-evaluating emerging signals across dimensions such as number of target events, lower limit of ROR, mortality proportion, criteria for important or designated medical events, and biological plausibility—provides a valuable framework. However, as the FAERS database does not provide sufficiently detailed mortality information for gastrointestinal adverse events associated with tirzepatide, our ability to perform a complete prioritization assessment was limited.

To address this, we have cited the relevant methodological literature (PMID: 39291217; 39972562) in the revised discussion section and explicitly acknowledged this limitation of our study (lines 349-351). We believe this addition helps place our findings into context while highlighting opportunities for future research using data sources that allow for a more comprehensive prioritization analysis.

Reviewer #3:

Comment 1: The authors should provide more detailed search strategies and screening flowcharts in the bibliometric analysis section. The current methodological description is overly simplified and lacks standardized systematic review processes. Particularly regarding keyword selection and database limitations, the authors could consider expanding the search scope to include other important biomedical databases such as PubMed and Embase to ensure comprehensive literature retrieval. Additionally, the authors should provide detailed descriptions of inclusion and exclusion criteria for literature screening and present inter-rater agreement results from two independent reviewers.

Response 1:

We sincerely thank the reviewer for this valuable suggestion. The topic of adverse events associated with tirzepatide spans multiple disciplines, including Clinical Medicine, Pharmacology, Biochemistry, and Public Health, thereby necessitating the use of a comprehensive database to ensure broad coverage. In the present study, we employed the Web of Science (WoS) Core Collection, which is widely recognized for its rigorous indexing standards and extensive inclusion of high-quality peer-reviewed journals. Its structure, grounded in Bradford’s Law and Garfield’s Law, increases the likelihood of capturing core publications while minimizing omissions. This choice is consistent not only with several recent high-quality bibliometric studies [1, 2] but also with a bibliometric analysis published by the academic editor of our manuscript [3-6], who similarly adopted a single-database approach. A detailed rationale for this selection has been provided in the Methods section (lines 78-82). Nevertheless, we fully acknowledge the limitation of relying on a single database, which may have excluded studies indexed exclusively in PubMed, Embase, or other biomedical resources. To address this concern, we have explicitly discussed this issue in the manuscript (lines 336-340), while emphasizing that the extensive coverage of WoS makes it unlikely that such omissions have substantially influenced the overall trends observed (lines 336-340).

Our search, conducted on November 18, 2024, used ‘tirzepatide’ and ‘adverse events’ as the primary keywords, supplemented with synonyms to maximize retrieval, which yielded 110 articles and reviews. The complete search strategy and bibliometric records are presented in the Supplementary Materials (S1 Table). It is important to note that bibliometric analysis differs from systematic reviews or meta-analyses. Specifically, bibliometric tools (e.g., VOSviewer, CiteSpace, biblioshiny) automatically remove duplicate and irrelevant records during analysis, thereby minimizing the need for manual screening. For this reason, we did not apply detailed manual inclusion and exclusion criteria or conduct dual independent screening with inter-rater agreement testing, which are methodological steps typically required in systematic reviews. Our approach is consistent with the bibliometric review recommended by the reviewer for our reference (doi:10.1016/j.cpan.2024.12.001).

Comment 2: In the FAERS data analysis, the authors should consider adopting more rigorous statistical methods to address potential confounding factors. While the current disproportionality analysis employed reporting odds ratios, it did not consider other algorithms such as IC method, PRR method, etc. Furthermore, the authors should also incorporate the VigiBase database for additional supplementary analysis (doi: 10.1016/j.eclinm.2024.102684).

Response 2: Thank you very much for this valuable suggestion. The Reporting Odds Ratio (ROR) is a well-established and widely utilized disproportionality method in pharmacovigilance research. Notably, it has served as the sole analytical approach in a number of high-quality studies, including the article kindly recommended by the reviewer (doi: 10.1016/j.eclinm.2024.102684). Based on its broad acceptance and support in the literature, we selected ROR as the primary analytical approach in the present study. Nevertheless, in order to further enhance the robustness of our results and to fully address the reviewer’s concern, we have additionally conducted complementary analyses using the Information Component (IC) and the Proportional Reporting Ratio (PRR) (lines 134-136). The findings from these supplementary analyses were consistent with our initial results (lines 207-208) and have now been included in the revised supplementary material (S7-9 Table).

With regard to the suggestion of incorporating VigiBase, we sincerely acknowledge its important value as a comprehensive global pharmacovigilance database. Although it was not included in the present study, we agree that integrating high-quality resources such as VigiBase in future research would further enhance the robustness and generalizability of our findings. This point has also been discussed in the limitations section of the revised manuscript (352-360).

Comment 3: The authors need to further explore the dose-response relationship and time-dependent characteristics of tirzepatide gastrointestinal adverse events. The current analysis primarily focuses on the frequency and distribution of adverse events but lacks in-depth analysis of event severity, duration, and clinical management strategies. Recent molecular mechanism research on adverse drug reactions shows that gastrointestinal adverse events are often closely related to drug target specificity and signal transduction pathways. The authors should combine tirzepatide's dual GIP/GLP-1 receptor agonist mechanism to deeply analyze the immune microenvironmental basis of its gastrointestinal effects (PMID: 35331128; 37691196; doi: 10.1002/mdr2.70001; 10.1002/mdr2.70007).

Response 3:

We sincerely thank the reviewer for this valuable suggestion. We fully agree that investigating the dose-response relationship, time-dependent characteristics, and clinical management strategies of tirzepatide-associated gastrointestinal adverse events would provide important insights. Unfortunately, the FAERS database, as a spontaneous reporting system, does not provide detailed information on drug dosage, treatment duration, event severity, or clinical management. Therefore, such analyses could not be performed within the scope of the present study. To our knowledge, other high-quality studies utilizing the FAERS database have also not been able to address these aspects due to the same inherent data limitations[7-12]. We have carefully acknowledged these limitations in the revised manuscript (lines 340-348) and highlighted that future prospective or real-world cohort studies with more granular data will be essential to address these important aspects (lines 352-360).

Regarding the suggested references (PMID: 35331128; 37691196; doi: 10.1002/mdr2.70001; 10.1002/mdr2.70007), we carefully reviewed them and found that they primarily focus on cancer research rather than the immune microenvironmental mechanisms of gastrointestinal adverse drug reactions. While these works are informative in their respective fields, their relevance to the mechanism of tirzepatide-related gastrointestinal effects may be limited. In the discussion, we discussed the possibility that the dual GIP/GLP-1 receptor agonist mechanism of tirzepatide may represent a plausible pharmacological basis for gastrointestinal adverse events (lines 313-323), supported by relevant literature[13, 14].We believe these clarifications and revisions improve the transparency of our study and appropriately delineate the scope and limitations of FAERS-based analyses.

Comment 4: Although the authors conducted subgroup analyses based on gender, age, and indications, they should further consider the impact of other important clinical variables on gastrointestinal adverse event incidence, such as comorbidities, hepatic and renal function status, and racial differences. Particularly in elderly patient populations, polypharmacy and organ function decline may significantly affect drug safety characteristics. The authors could employ multivariate regression analysis or propensity score matching methods to more accurately assess the impact of these confounding factors.

Response 4:

We sincerely appreciate the reviewer’s insightful suggestion. We fully agree that incorporating additional clinical variables-such as comorbidities, hepatic and renal function status, racial differences, and polypharmacy in elderly patients-would provide valuable insights into the safety profile of tirzepatide. Unfortunately, the FAERS database, being a spontaneous reporting system, does not capture such detailed clinical information. As a result, it is not feasible to perform multivariate regression analyses, propensity score matching, or other advanced statistical adjustments within the scope of this study. This inherent limitation has also been noted in other high-quality FAERS-based studies[7-12]. We have highlighted this issue in the revised manuscript (lines340-348) and emphasized that future prospective studies or real-world cohort investigations with more comprehensive clinical data will be essential to address these important aspects (lines 352-360).

Comment 5: The two analytical components in the current study are relatively independent, lacking deep integration. The authors should establish correlation analyses between literature research hotspots and real-world adverse event reports, exploring differences between academic research focus and actual safety issues occurring in clinical practice. This integrated analysis is of significant importance for guiding future research directions and clinical monitoring priorities. With the widespread application of multi-omics technologies in drug discovery, prediction and mechanistic research of adverse drug reactions also require more systematic methodological support (doi: 10.1016/j.cpan.2024.12.001).

Response 5:

We sincerely thank the reviewer for this valuable suggestion. We fully agree that integrating bibliometric analysis with real-world pharmacovigilance data could provide deeper insights into the alignment and discrepancies between academic research priorities and clinical safety signals. In fact, our discussion section has touched upon this aspect: while bibliometric analysis highlighted cardiovascular events as a major research hotspot, our FAERS-based pharmacovigilance assessment revealed gastrointestinal adverse events (GIAEs) as the predominant safety concern reported in clinical practice. This contrast underscores the current research gap regarding tirzepatide-associated GIAEs and highlights the importance of further investigation in this area (lines 271-283).

we agree with the reviewer that future research would benefit from more systematic methodological frameworks. In particular, combining multi-omics approaches, predictive modeling, and real-world safety monitoring has great potential to bridge the gap between research focus and clinical needs.

Comment 6: The authors should more comprehensively discuss the inherent limitations of the FAERS database, including reporting bias, missing information, limitations of causal inference, and propose corresponding solutions or explanatory approaches. Additionally, they need to explain how to overcome these limitations in actual clinical applications to improve the practical utility of research results.

Response 6:

We sincerely thank the reviewer for this constructive suggestion. We fully agree that a more comprehensive discussion of the inherent limitations of the FAERS database, as well as potential solutions, is essential for improving the interpretability and clinical utility of our findings. In the revised manuscript, we have expanded the discussion of these limitations (lines 340-352). Specifically, we emphasized that FAERS, as a spontaneous reporting system, is subject to underreporting, reporting bias, incomplete data, and potential duplicate records. Moreover, it lacks detailed clinical variables such as drug dosage, treatment duration, event severity, comorbidities, and organ function status, which limits the ability to explore causal relationships or conduct advanced statistical adjustments.

To address these issues, we highlighted several strategies: (1) integrating pharmacovigilance data with prospective cohort studies or electronic health record databases to obtain more granular clinical information; (2) applying complementary methodologies such as active surveillance systems, signal detection algorithms, and mechanistic studies to strengthen causal inference; and (3) encouraging standardized clinical reporting to reduce missing data and reporting bias. From a clinical perspective, overcoming these limitations requires careful interpretation of FAERS signals in conjunction with evidence from randomized controlled trials and real-world cohort studies, thereby improving the applicability of pharmacovigilance findings to patient care.

We have incorporated these points into the revised discussion and conclusion to clarify both the constraints of our study and the potential pathways for enhancing the practical value of FAERS-based research on tirzepatide (lines 352-360).

References

1. Wu T, Wu Y, Nie K, Yan J, Chen Y, Wang S, et al. Bibliometric analysis and global trends in uterus transplantation. Int J Surg. 2024;110(8):4932-46. Epub 202408

---

## [Decision Letter · Decision Letter 2]

25 Nov 2025

Dear Dr. Kwang Joon Kim,

Thank you for submitting your manuscript to PLOS ONE. After careful consideration, we feel that it has merit but does not fully meet PLOS ONE’s publication criteria as it currently stands. Therefore, we invite you to submit a revised version of the manuscript that addresses the points raised during the review process.

**ACADEMIC EDITOR:** Prior to further processing, the author is required to consider the reviewers' suggestions and make revisions accordingly.

We look forward to receiving your revised manuscript.

Kind regards,

Jingyu Wang

Academic Editor

PLOS ONE

Journal Requirements:

Additional Editor Comments (if provided):

Prior to further processing, the authors are required to consider the reviewers' suggestions and make revisions accordingly.

Reviewers' comments:

Reviewer's Responses to Questions

**Comments to the Author**

Reviewer #3: (No Response)

Reviewer #4: (No Response)

2. Is the manuscript technically sound, and do the data support the conclusions?

Reviewer #3: (No Response)

Reviewer #4: Yes

3. Has the statistical analysis been performed appropriately and rigorously?

Reviewer #3: (No Response)

Reviewer #4: Yes

4. Have the authors made all data underlying the findings in their manuscript fully available?

Reviewer #3: (No Response)

Reviewer #4: Yes

5. Is the manuscript presented in an intelligible fashion and written in standard English?

Reviewer #3: (No Response)

Reviewer #4: Yes

Reviewer #3: Thank you for the comprehensive revisions to your manuscript. I am pleased to note that the majority of the suggested modifications have been successfully addressed, and the overall quality of the work has been significantly improved. However, there is one remaining area that would benefit from further expansion and discussion. With the widespread application of multi-omics technologies in drug discovery, prediction and mechanistic research of adverse drug reactions also require more systematic methodological support (doi: 10.1016/j.cpan.2024.12.001; 10.1016/j.cpan.2024.12.002). Also, authors need to further explore the dose-response relationship and time-dependent characteristics of tirzepatide gastrointestinal adverse events. The current analysis primarily focuses on the frequency and distribution of adverse events but lacks in-depth analysis of event severity, duration, and clinical management strategies. Recent molecular mechanism research on adverse drug reactions shows that gastrointestinal adverse events are often closely related to drug target specificity and signal transduction pathways. The authors should combine tirzepatide's dual GIP/GLP-1 receptor agonist mechanism to deeply analyze the immune microenvironmental basis of its gastrointestinal effects (PMID: 35331128; 37691196; doi: 10.1002/mdr2.70001; 10.1002/mdr2.70007).

Reviewer #4: 1. Overall Assessment:

This study integrates bibliometric and FAERS pharmacovigilance analyses to evaluate gastrointestinal adverse events (GIAEs) associated with tirzepatide. The topic is clinically relevant and fills a gap in real-world safety data. However, substantial methodological and structural improvements are required before acceptance.

2. Major Comments:

2.1 Bibliometric Methodology Needs More Standardization:

The rationale for choosing WoSCC is explained, but the search strategy requires a complete Boolean expression, clearer PRISMA-style workflow, and explicit inclusion/exclusion criteria.

2.2 FAERS Analysis Requires More Detailed Standardization:

Clarify duplicate handling, primary vs. secondary suspect definitions, and missing data treatment. Consider adding serious outcome indicators.

2.3 Results Section Contains Redundant Figures:

Figures (e.g., multiple Venn diagrams) may overwhelm readers. Consolidation into heatmaps or upset plots is recommended.

2.4 Discussion Section Should Strengthen Mechanistic Interpretation:

Deepen the explanation of eructation, delayed gastric emptying, sex differences, and T2DM-related gastric motility issues.

2.5 Add a 'Clinical Implications' Summary:

Highlight dose titration strategies, monitoring recommendations, and subgroup-specific safety considerations.

2.6 Integration Between Bibliometric and FAERS Sections Is Weak:

Provide an integrated visual summary and analysis of discrepancies between research hotspots and real-world AE signals.

3. Minor Comments:

3.1 Language Requires Polishing.

3.2 Table Formatting Should Be Standardized.

3.3 Figure Legends Should Follow PLOS ONE Conventions.

3.4 Verify Reference Formatting and Consistency.

3.5 Add Software Versions Used in Analysis.

The most important point is the lack of cutting-edge literature. Please refer to the following references: PMID: 40904688, PMID: 40349689, PMID: 38229552, PMID: 39581182, PMID: 40142667

**Do you want your identity to be public for this peer review?** For information about this choice, including consent withdrawal, please see our For information about this choice, including consent withdrawal, please see our Privacy Policy .

Reviewer #3: No

Reviewer #4: **Yes:** Qian GuoQian Guo

---

## [Author Response · Author response to Decision Letter 3]

8 Jan 2026

Response to Editor and Reviewers

Dear Editor,

We sincerely thank the reviewers for their insightful comments and constructive suggestions, which have substantially improved the quality and clarity of our manuscript.

We have carefully revised the manuscript in response to the reviewers’ feedback and resubmitted it for your consideration. All revisions are highlighted in yellow in the revised version, and a detailed, point-by-point response to each comment is provided below.

We hope that the revised manuscript meets the requirements for publication in the journal and appreciate your time and consideration. We look forward to your response.

With best regards,

Corresponding Author:

Kwang Joon Kim, PhD

College of Pharmacy, Chonnam National University, Gwangju 61186, Republic of Korea

Phone: 82 625302929

Fax: 82 625302949

Email: kjkim0901@jnu.ac.kr

Co-corresponding Author

Kyung-In Joung, PhD

School of AI Healthcare, College of Integrated Health Science, CHA University, Pocheon-si 11160, Republic of Korea

Phone: 82 318509087

Fax: 82 315439917

Email: jki0515@cha.ac.kr

Detailed responses to the reviewers’ comments and suggestions are provided below.

Reviewer #3:

Comment 1

With the widespread application of multi-omics technologies in drug discovery, prediction, and mechanistic research of adverse drug reactions also requires more systematic methodological support (doi: 10.1016/j.cpan.2024.12.001; 10.1016/j.cpan.2024.12.002).

Response 1

Thank you for this valuable suggestion. We have added a concise statement to the Discussion highlighting the relevance of multi-omics and network pharmacology approaches and have cited the recommended references ([46,47]; doi: 10.1016/j.cpan.2024.12.001; 10.1016/j.cpan.2024.12.002; lines 271–276).

Comment 2

Also, authors need to further explore the dose-response relationship and time-dependent characteristics of tirzepatide gastrointestinal adverse events. The current analysis primarily focuses on the frequency and distribution of adverse events but lacks in-depth analysis of event severity, duration, and clinical management strategies. Recent molecular mechanism research on adverse drug reactions shows that gastrointestinal adverse events are often closely related to drug target specificity and signal transduction pathways. The authors should combine tirzepatide's dual GIP/GLP-1 receptor agonist mechanism to deeply analyze the immune microenvironmental basis of its gastrointestinal effects (PMID: 35331128; 37691196; doi: 10.1002/mdr2.70001; 10.1002/mdr2.70007)

Response 2

Thank you for this insightful comment. In response, we have expanded the Discussion to (1) acknowledge FAERS limitations regarding dose–response assessment, severity grading, treatment duration, and clinical management (lines 295–299), and (2) provide a more detailed mechanistic interpretation of tirzepatide-related GIAEs. We have also incorporated the reviewer-recommended literature to support the expanded discussion ([48–51]; PMID: 35331128, 37691196; doi: 10.1002/mdr2.70001, 10.1002/mdr2.70007; (lines 276–282).

Reviewer #4:

Comment 2.1

Bibliometric Methodology Needs More Standardization: The rationale for choosing WoSCC is explained, but the search strategy requires a complete Boolean expression, clearer PRISMA-style workflow, and explicit inclusion/exclusion criteria.

Response 2.1

We appreciate the reviewer’s suggestion regarding methodological standardization in bibliometric research and agree that transparency in data retrieval is essential. We clarify that the complete Boolean search strategy, rationale for selecting the WoSCC, and data preprocessing procedures were fully reported in the original submission. We have also re-examined the 110 included studies for duplicate publications and identified none.

Regarding the request for a PRISMA-style workflow and explicit inclusion and exclusion criteria, we have clarified the methodological rationale of this study. Our analysis is grounded in science mapping theory and implemented using the bibliometrix R package, which provides a standardized, well-documented bibliometric workflow [1] and conceptually aligns with the CiteSpace framework developed by Chen [2, 3]. These approaches conceptualize scientific literature as citation- and co-occurrence–based networks to identify knowledge structures, research fronts, and their temporal evolution, rather than to synthesize evidence or estimate effect sizes.

As emphasized in Chen’s foundational work [2, 3], bibliometric mapping aims to preserve the structural integrity and global connectivity of citation networks to comprehensively represent the intellectual landscape of a research domain. Within this framework, both highly cited core publications and less-cited peripheral or intermediary studies contribute to co-citation structures, thematic clusters, and their temporal evolution.

Applying restrictive inclusion and exclusion criteria similar to those used in systematic reviews or meta-analyses may remove publications that, despite low citation counts or apparent topical marginality, serve as intellectual bridges between otherwise disconnected research fronts. Excluding these bridging nodes can fragment co-citation networks, alter cluster boundaries, and bias the identification of emerging topics or evolutionary pathways, resulting in a distorted or incomplete representation of knowledge evolution over time.

Accordingly, and consistent with established bibliometric practice and the methodological framework of the bibliometrix R package [15], this study does not apply PRISMA-style screening procedures, which are designed for evidence synthesis and causal inference. For transparency and verification, the search strategy and data processing details are reported in the Methods section (lines 76–85) and S1 Table.

Comment 2.2

FAERS Analysis Requires More Detailed Standardization: Clarify duplicate handling, primary vs. secondary suspect definitions, and missing data treatment. Consider adding serious outcome indicators.

Response 2.2

We appreciate the reviewer’s valuable comments regarding FAERS analysis standardization. In this study, the FAERS data processing strictly followed the official guidelines provided on the FDA FAERS website. The procedures are detailed in the Methods section (lines 96–102), and Supplementary Figure S1 shows the complete workflow. Our approach is also fully consistent with the methodological recommendations of the reviewer-suggested studies ([18,19,30]: PMID 40904688, 40349689, 40142667), which have been cited in the revised Methods section (line 101).

In FAERS, each drug is assigned a standardized role indicating its presumed involvement in an adverse event. Primary suspect drugs are considered most likely to have caused the event, whereas secondary suspects indicate a possible but less certain association. Consistent with the reviewer-recommended studies cited in our manuscript (line 101) ([18,19,30]), we restricted signal-detection analyses to primary suspect drugs to reduce potential confounding, as described in the Methods section (lines 93-94).

Finally, because FAERS lacks sufficiently detailed mortality information (serious outcome indicators) related to tirzepatide-associated gastrointestinal adverse events, a comprehensive severity assessment was not feasible. This limitation is explicitly acknowledged in the manuscript (lines 300–301).

Comment 2.3

Results Section Contains Redundant Figures: Figures (e.g., multiple Venn diagrams) may overwhelm readers. Consolidation into heatmaps or upset plots is recommended.

Response to Comment 2.3

We thank the reviewer for the constructive suggestion to address potential redundancy in the Results figures and for recommending the use of UpSet plots.

In response, we have generated UpSet plots for the overall population as well as sex- and age-stratified groups, and presented them here for the reviewer’s reference. This allowed a direct comparison of the effectiveness of different visualization approaches.

Our comparison showed that, for analyses involving a small number of sets (three per comparison), the original Venn diagrams (Figure 3B & C) provide a more intuitive visualization of shared and unique gastrointestinal adverse event (GIAE) signals across subgroups. These diagrams enable rapid comprehension of overlap patterns that are central to the Results. The accompanying heatmaps (Figures 3D & E) then add complementary quantitative detail by depicting signal strength and subgroup-specific differences. Collectively, these figures offer a clear and logically progressive presentation rather than redundant information.

Although the UpSet plots accurately depict the same intersections, they did not enhance the interpretability of the key findings in this context and appeared unnecessarily complex given the limited number of sets. We therefore retained the original Venn diagrams and heatmaps, which provide the clearest and most accessible presentation of the results.

Nevertheless, we acknowledge the value of UpSet plots for visualizing complex intersections involving larger numbers of sets and appreciate the reviewer’s suggestion, which prompted us reassess and confirm the appropriateness of the final figure design.

Comment 2.4

Discussion Section Should Strengthen Mechanistic Interpretation:

Deepen the explanation of eructation, delayed gastric emptying, sex differences, and T2DM-related gastric motility issues.

Response 2.4

Thank you for this insightful comment. In response, we have substantially strengthened the mechanistic interpretation in the Discussion section. Specifically, we expanded the discussion of mechanisms underlying eructation and delayed gastric emptying, and provided a more detailed explanation of sex-related differences and T2DM-associated alterations in gastric motility (lines 260–270).

Comment 2.5

Add a 'Clinical Implications' Summary: Highlight dose titration strategies, monitoring recommendations, and subgroup-specific safety considerations.

Response 2.5

We appreciate your suggestion. In the revised manuscript, we have added a dedicated Clinical Implications subsection to the Discussion, focusing on clinically actionable recommendations supported by our data.

These include emphasizing close monitoring during the first 3 months of treatment, when most gastrointestinal adverse events occur, and identifying higher-risk subgroups, such as older adults, males, patients receiving concomitant medications, and patients with T2DM (lines 309–314). We have also revised the Conclusion to align with these additions and to avoid redundancy with the new Clinical Implications subsection (lines 30–33,315–318).

Comment 2.6

Integration Between Bibliometric and FAERS Sections Is Weak: Provide an integrated visual summary and analysis of discrepancies between research hotspots and real-world AE signals.

Response 2.6

We appreciate the reviewer’s insightful comment and agree that better integration of the bibliometric findings with the FAERS pharmacovigilance results would enhance the coherence of the manuscript. In response, we have made substantial revisions to the Discussion.

Specifically, we clarified why cardiovascular outcomes emerged as the dominant research hotspot in the bibliometric analysis and why the investigation of GIAEs remains clinically important despite this emphasis (lines 237–247). We further explained that cardiovascular outcomes have historically received disproportionate attention because T2DM markedly increases the risk of major cardiovascular events and because regulatory agencies require dedicated cardiovascular outcomes trials for new antidiabetic therapies, which collectively drive the literature toward cardiovascular themes.

In contrast, tirzepatide-related GIAEs, although underrepresented in the published literature, emerge as prominent and disproportionate signals in real-world FAERS data. The expanded Discussion highlights this discrepancy between prevailing research priorities and patient-reported safety concerns. By integrating bibliometric and pharmacovigilance findings, we underscore an unmet need for systematic evaluation of tirzepatide-associated GIAEs, an understudied issue with substantial implications for clinical decision-making, patient adherence, and risk–benefit assessment.

To improve structural coherence, we have added a dedicated paragraph in the Methods section describing the integrated study design that combines bibliometric analysis and pharmacovigilance assessment, thereby clarifying how the two components were jointly conceptualized and analyzed (lines 68–73).

We have also added a new figure (Fig. 1) providing an overview of the study design and the integration of bibliometric and FAERS analyses, which visually demonstrates how the two components complement each other and facilitates interpretation of discrepancies between research hotspots and real-world adverse event signals (line 73).

Comment 3.1

Language Requires Polishing.

Response 3.1

We thank the reviewer for this constructive comment. The manuscript has been thoroughly revised to improve clarity, grammatical accuracy, and academic tone (as highlighted in yellow). In addition, the manuscript underwent professional language editing, and the corresponding certificate is provided as an attachment. All revisions were made to enhance readability while preserving the original scientific meaning.

Comment 3.2

Table Formatting Should Be Standardized.

Response 3.2

Thank you for this valuable comment. The tables have been carefully reviewed and reformatted to ensure consistency and compliance with the journal’s requirements (see Supporting Information Table

Comment 3.3

Figure Legends Should Follow PLOS ONE Conventions.

Response 3.3

We thank the reviewer for this helpful comment. All figure legends have been revised to comply with PLOS ONE formatting conventions (lines 155–159, 184-188, 198-201,214-219,232-233).

Comment 3.4

Verify Reference Formatting and Consistency.

Response 3.4

Thank you for this important comment. All references have been carefully checked.

Comment 3.5

Add Software Versions Used in Analysis.

Response 3.5

Thank you for this helpful suggestion. We have revised the Methods section to explicitly report the software versions used. All bibliometric and pharmacovigilance analyses were conducted in R (version 4.4.2), with bibliometric analyses performed using the bibliometrix package and FAERS data processing and visualization conducted using easyFAERS and ggplot2, respectively. These details have been added to the revised manuscript (lines 82-83, 105–106).

Comment 3.6

The most important point is the lack of cutting-edge literature. Please refer to the following references: PMID: 40904688, PMID: 40349689, PMID: 38229552, PMID: 39581182, PMID: 40142667

Response 3.6

We thank the reviewer for highlighting the importance of including recent literature. We have carefully reviewed the suggested references ([18,19,21,24,30]; PMID: 40904688, 40349689, 38229552, 39581182, 40142667) and incorporated relevant information into the Methods section (lines 63,79, 101). These additions contextualize our study within the recent advances in the field and strengthen its scientific foundation.

References

1. Aria M, Cuccurullo C. bibliometrix: An R-tool for comprehensive science mapping analysis. Journal of Informetrics. 2017;11(4):959-75.

2. Chen C. CiteSpace II: Detecting and visualizing emerging trends and transient patterns in scientific literature. Journal of the American Society for Information Science and Technology. 2006;57(3):359-77.

3. Chen C. CiteSpace: a practical guide for mapping scientific literature: Nova Science Publishers Hauppauge, NY, USA; 2016.

1. Aria M, Cuccurullo C. bibliometrix: An R-tool for comprehensive science mapping analysis. Journal of informetrics. 2017;11(4):959-75.

2. Chen C. CiteSpace II: Detecting and visualizing emerging trends and transient patterns in scientific literature. Journal of the American Society for information Science and Technology. 2006;57(3):359-77.

3. Chen C. CiteSpace: a practical guide for mapping scientific literature: Nova Science Publishers Hauppauge, NY, USA; 2016.

---

## [Decision Letter · Decision Letter 3]

27 Feb 2026

Dear Dr. Kim,

Thank you for submitting your manuscript to PLOS ONE. After careful consideration, we feel that it has merit but does not fully meet PLOS ONE’s publication criteria as it currently stands. Therefore, we invite you to submit a revised version of the manuscript that addresses the points raised during the review process.

We look forward to receiving your revised manuscript.

Kind regards,

Jingyu Wang

Academic Editor

PLOS One

Journal Requirements:

Reviewers' comments:

Reviewer's Responses to Questions

**Comments to the Author**

Reviewer #3: (No Response)

Reviewer #4: All comments have been addressed

2. Is the manuscript technically sound, and do the data support the conclusions?

Reviewer #3: (No Response)

Reviewer #4: Yes

3. Has the statistical analysis been performed appropriately and rigorously?

Reviewer #3: (No Response)

Reviewer #4: Yes

4. Have the authors made all data underlying the findings in their manuscript fully available?

Reviewer #3: (No Response)

Reviewer #4: Yes

5. Is the manuscript presented in an intelligible fashion and written in standard English?

Reviewer #3: (No Response)

Reviewer #4: Yes

Reviewer #3: The authors have satisfactorily addressed several of the previous concerns. However, regarding the references, I noticed that citations 46, 47, 48, 49, and 51 do not appear to be well-suited for the specific context of this article. I suggest that the authors remove these references to ensure the bibliography is strictly relevant to the study.

Reviewer #4: (No Response)

**Do you want your identity to be public for this peer review?** For information about this choice, including consent withdrawal, please see our For information about this choice, including consent withdrawal, please see our Privacy Policy .

Reviewer #3: No

Reviewer #4: No

---

## [Author Response · Author response to Decision Letter 4]

3 Mar 2026

We have provided our detailed responses to the reviewers’ comments and suggestions as follows.

Reviewer #3:

Comment 1

Review Comments to the Author

Reviewer #3: The authors have satisfactorily addressed several of the previous concerns. However, regarding the references, I noticed that citations 46, 47, 48, 49, and 51 do not appear to be well-suited for the specific context of this article. I suggest that the authors remove these references to ensure the bibliography is strictly relevant to the study.

Response 1

We sincerely thank the reviewer for the careful evaluation of our reference list. Following the reviewer’s suggestion, we have removed them from the revised version and have cited more appropriate references in their place (Lines 276–279).

Reviewer #4:

Comment 1

Review Comments to the Author

Reviewer #4: (No Response)

Response 1

We thank the reviewer for the overall positive evaluation and constructive feedback provided throughout the review process

---

## [Decision Letter · Decision Letter 4]

13 Mar 2026

Gastrointestinal adverse events associated with tirzepatide: a bibliometric and pharmacovigilance analysis

PONE-D-25-13689R4

Dear Dr. Kwang Joon Kim,

We’re pleased to inform you that your manuscript has been judged scientifically suitable for publication and will be formally accepted for publication once it meets all outstanding technical requirements.

Kind regards,

Jingyu Wang

Academic Editor

PLOS One

Additional Editor Comments (optional):

Reviewers' comments:

Reviewer's Responses to Questions

**Comments to the Author**

Reviewer #3: (No Response)

Reviewer #4: All comments have been addressed

2. Is the manuscript technically sound, and do the data support the conclusions?

Reviewer #3: (No Response)

Reviewer #4: Yes

3. Has the statistical analysis been performed appropriately and rigorously?

Reviewer #3: (No Response)

Reviewer #4: Yes

4. Have the authors made all data underlying the findings in their manuscript fully available?

Reviewer #3: (No Response)

Reviewer #4: Yes

5. Is the manuscript presented in an intelligible fashion and written in standard English?

Reviewer #3: (No Response)

Reviewer #4: Yes

Reviewer #3: The authors have satisfactorily addressed the previous queries. This version of manuscript is acceptable for the journal.

Reviewer #4: (No Response)

**Do you want your identity to be public for this peer review?** For information about this choice, including consent withdrawal, please see our For information about this choice, including consent withdrawal, please see our Privacy Policy .

Reviewer #3: No

Reviewer #4: No

---

## [Editor Report · Acceptance letter]

PONE-D-25-13689R4

PLOS One

Dear Dr. Kim,

I'm pleased to inform you that your manuscript has been deemed suitable for publication in PLOS One. Congratulations! Your manuscript is now being handed over to our production team.

Kind regards,

on behalf of

Dr. Jingyu Wang

Academic Editor

PLOS One